# COMMUTE GRAPH NEURAL NETWORKS

## ABSTRACT

Graph Neural Networks (GNNs) have shown remarkable success in learning from graph-structured data. However, their application to directed graphs (digraphs) presents unique challenges, primarily due to the inherent asymmetry in node relationships. Traditional GNNs are adept at capturing unidirectional relations but fall short in encoding the mutual path dependencies between nodes, such as asymmetrical shortest paths typically found in digraphs. Recognizing this gap, we introduce Commute Graph Neural Networks (CGNN), an approach that seamlessly integrates node-wise commute time into the message passing scheme. The cornerstone of CGNN is an efficient method for computing commute time using a newly formulated digraph Laplacian. Commute time is then integrated into the neighborhood aggregation process, with neighbor contributions weighted according to their respective commute time to the central node in each layer. It enables CGNN to directly capture the mutual, asymmetric relationships in digraphs. Extensive experiments confirm the superior performance of CGNN. Source code of CGNN is anonymously available `here`.

## 1 INTRODUCTION

Directed graphs (digraphs) are widely employed to model relational structures in diverse domains, such as social networks (Cross et al., 2001) and recommendation systems (Qiu et al., 2020). Recently, the advances of graph neural networks (GNNs) have inspired various attempts to adopt GNNs for analyzing digraphs (Tong et al., 2020a;b; 2021; Zhang et al., 2021; Rossi et al., 2023; Geisler et al., 2023). The essence of GNN-based digraph analysis lies in utilizing GNNs to learn expressive node representations that encode edge direction information.

To achieve this, modern digraph neural networks are designed to integrate edge direction information into the message passing process by distinguishing between incoming and outgoing edges. This distinction enables the central node to learn directionally discriminative information from its neighbors. As illustrated in the digraph of Fig. 1, given a central node $v_i$, a 1-layer digraph neural network can aggregate messages from $v_i$'s incoming neighbor $v_m$ and outgoing neighbor $v_j$, and simultaneously capture edge directions by applying direction-specific aggregation functions (Rossi et al., 2023), or by predefining edge-specific weights (Zhang et al., 2021; Tong et al., 2020b).

Despite the advancements, current digraph neural networks primarily capture unidirectional[1] relationships between nodes, neglecting the complexity arising from path asymmetry. For instance, a $k$-layer GNN aggregates the neighbors within the shortest path $k$ for the central node. If the graph is undirected, the shortest path between any two nodes is symmetric, as shown in the undirected graph of Fig. 1. This symmetry simplifies the representation of node relationships, implying that if the shortest path distances (SPDs) from one node to two other nodes are identical, then the SPDs from these two nodes back to the source node must also be the same. Conversely, such symmetry is absent in digraphs. Considering the digraph in Fig. 1, the shortest paths between $v_i$ and $v_j$ are asymmetric. Therefore, although $v_j$ and $v_k$ are both immediate outgoing neighbors of $v_i$, the strength of their relationships with the central node differs significantly. Existing methods (Rossi et al., 2023; Tong et al., 2020b; Zhang et al., 2021), by focusing solely on unidirectional shortest paths (blue and red arrows), fail to capture the asymmetry phenomenon, which conveys valuable information of node relationships. Take social networks as an example: an ordinary user can directly follow a celebrity,

---

[1] 'unidirectional' refers to relationships in digraphs where edges have a specific direction from one node to another.

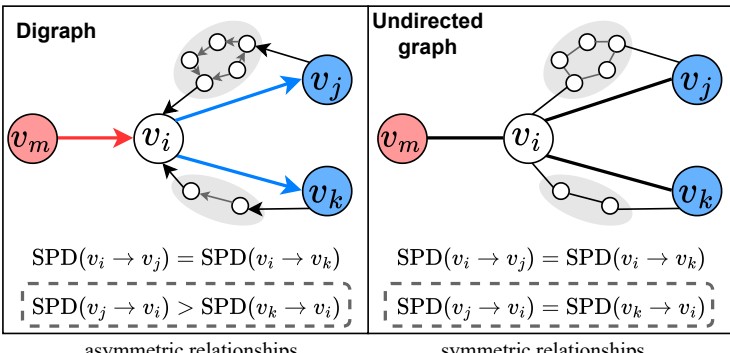

Figure 1: A digraph and its undirected counterpart. Blue arrows indicate unidirectional paths, together with longer paths in the gray area, forming commute closed loops between the central node $v_i$ and its outgoing neighbors $v_j$ and $v_k$. In the undirected graph, shortest path distances (SPD) between nodes are symmetric. However, in the digraph, the fact that unidirectional SPDs are equal does not imply that mutual SPDs will also be equal. For instance, while the SPDs from $v_i$ to $v_j$ and $v_k$ are identical, the reverse SPD from $v_j$ and $v_k$ back to $v_i$ do not necessarily match these distances.

yielding a short path to the celebrity, yet the reverse path from the celebrity to the follower might be much longer. Considering only the short path from the follower to celebrity could falsely suggest a level of closeness that does not exist. In contrast, accounting for the mutual paths between users yields a more precise and robust measure of their relationship, with stronger mutual interactions implying stronger connections.

To capture the mutual path interactions in GNNs, we adapt the concept of commute time, the expected number of steps to traverse from a source node to a target and back, from the Markov chain theory to the domain of graph learning. To this end, we first generalize the graph Laplacian to the digraph by defining the divergence of the gradient on the digraph. Utilizing this digraph-specific Laplacian, we develop an efficient method to compute commute time, ensuring sparsity and computational feasibility. Then we incorporate the commute-time-based proximity measure into the message passing process by assigning aggregation weights to neighbors. The intuition behind is that the immediate and unidirectional neighboring relationships do not necessarily imply strong similarity, but the mutual proximity is a more reliable indicator of relationship closeness. Our experimental results demonstrate the effectiveness of CGNN.

Our main contributions are as follows:

i. We identify and address mutual path dependencies in directed graphs, which is crucial for representing real-world relationships between entities, a factor ignored in prior work. Further, we propose to use commute times to quantify the strength of node-wise mutual path dependencies.

ii. We extend the traditional graph Laplacian to directed graphs by introducing `DiLap`, a novel Laplacian based on signal processing principles tailored for digraphs. Leveraging `DiLap`, we develop an efficient and theoretically sound method for computing commute times that enhances computational feasibility.

iii. We propose the Commute Graph Neural Networks (CGNN), which incorporate commute-time-weighted message passing into their architecture. Through comprehensive experiments across various digraph datasets, we demonstrate the effectiveness of CGNN.

## 2  PRELIMINARY

**Notations**   Consider $G = (V, E, \mathbf{X})$ as an unweighted digraph comprising $N$ nodes, where $V = \{v_i\}_{i=1}^N$ is the node set, $E \subseteq (V \times V)$ is the edge set with size $M$, $\mathbf{X} \in \mathbb{R}^{N \times d}$ is the node feature matrix. $Y = \{y_1, \cdots, y_N\}$ is the set of labels for $V$. Let $\mathbf{A} \in \mathbb{R}^{N \times N}$ be the adjacency matrix and $\mathbf{D} = \mathrm{diag}(d_1, \cdots, d_N) \in \mathbb{R}^{N \times N}$ be the degree matrix of $\mathbf{A}$, where $d_i = \sum_{v_j \in V} \mathbf{A}_{ij}$ is the out-degree of $v_i$. Let $\tilde{\mathbf{A}} = \mathbf{A} + \mathbf{I}$ and $\tilde{\mathbf{D}} = \mathbf{D} + \mathbf{I}$ denote the adjacency and degree matrix with self-loops, respectively. The transition probability matrix of the Markov chain associated with random walks on

$G$ is defined as $\mathbf{P} = \mathbf{D}^{-1}\mathbf{A}$, where $\mathbf{P}_{ij} = \mathbf{A}_{ij}/\deg(v_i)$ is the probability of a 1-step random walk starting from $v_i$ to $v_j$. Given that $\mathbf{D}^{-1}$ is a diagonal matrix and considering that real-world graphs are typically sparse ($M \ll N^2$), $\mathbf{A}$ and consequently $\mathbf{P}$ can generally be considered sparse. Graph Laplacian formalized as $\mathbf{L} = \mathbf{D} - \mathbf{A}$ is defined on the undirected graph whose adjacency matrix is symmetric. The symmetrically normalized Laplacian with self-loops (Wu et al., 2019) is defined as $\hat{\mathbf{L}} = \tilde{\mathbf{D}}^{-\frac{1}{2}}\tilde{\mathbf{L}}\tilde{\mathbf{D}}^{-\frac{1}{2}}$, where $\tilde{\mathbf{L}} = \tilde{\mathbf{D}} - \tilde{\mathbf{A}}$.

**Digraph Neural Networks**  DirGNN (Rossi et al., 2023) is a general framework that generalizes the message passing paradigm to digraphs by adapting to the directionality of edges. It involves separate aggregation processes for incoming and outgoing neighbors of each node as follows:

$$
\begin{aligned}
m_{i,\text{in}}^{(\ell)} &= \text{Agg}_{\text{in}}^{(\ell)} \left( \left\{ h_j^{(\ell-1)} : v_j \in \mathcal{N}_i^{\text{in}} \right\} \right) \\
m_{i,\text{out}}^{(\ell)} &= \text{Agg}_{\text{out}}^{(\ell)} \left( \left\{ h_j^{(\ell-1)} : v_j \in \mathcal{N}_i^{\text{out}} \right\} \right) \\
h_i^{(l)} &= \text{Comb}^{(\ell)} \left( h_i^{(\ell-1)}, m_{i,\text{in}}^{(\ell)}, m_{i,\text{out}}^{(\ell)} \right),
\end{aligned}
\tag{1}
$$

where $\mathcal{N}_i^{\text{in}}$ and $\mathcal{N}_i^{\text{out}}$ are respectively incoming and outgoing neighbors of $v_i$. $\text{Agg}_{\text{in}}^{(\ell)}(\cdot)$ and $\text{Agg}_{\text{out}}^{(\ell)}(\cdot)$ are specialized aggregation functions of $\mathcal{N}_i^{\text{in}}$ and $\mathcal{N}_i^{\text{out}}$ at layer $\ell$, used to encode the directional characteristics of the edges connected to $v_i$.

## 3   RANDOM WALK DISTANCE AND GNNs

Based on the established notations, we then show that message passing based GNNs naturally capture the concept of hitting time during information propagation across the graph, due to the unidirectional nature of the neighborhood aggregation. Subsequently, we argue for the significance of commute time, highlighting it as a more compact measure of mutual node-wise interactions in random walks.

### 3.1   CAN GNNs CAPTURE RANDOM WALK DISTANCE?

In the context of random walks on a digraph, hitting time and commute time, collectively referred to as random walk distances, serve as key metrics for assessing node connectivity and interaction strength. Hitting time $h(v_i, v_j)$ is the expected number of steps a random walk takes to reach a specific target node $v_j$ for the first time, starting from a given source node $v_i$. Commute time $c(v_i, v_j)$ is the expected number of steps required for a random walk to start at $v_i$, reach $v_j$, and come back. A high hitting (commute) time indicates difficulty in achieving unidirectional (mutual) visits to each other in a random walk. As illustrated in the digraph of Fig. 1, commute time $c(v_i, v_j) > c(v_i, v_k)$, while the hitting time $h(v_m, v_i) = h(v_i, v_j) = h(v_i, v_k)$.

**Motivation**  Given these definitions, two questions arise: How crucial is it to retain these measures in graph learning? Also, are message-passing GNNs capable of preserving these characteristics? Firstly, both hitting time and commute time are critical in understanding the structural dynamics of graphs. Hitting time, analogous to the shortest path, measures the cost of reaching one node from another, reflecting the directed influence or connectivity. Commute time, encompassing the round-trip journey, offers insights into the mutual relationships between nodes, which is especially evident in social networks, as illustrated by celebrity-follower relationships. Secondly, message-passing GNNs are somewhat effective in capturing hitting time, as they propagate information across the graph in a manner similar to a random walk, where quickly reached nodes are preferentially aggregated, and the influence of nodes exponentially diminishes with increasing distance (Topping et al., 2022). However, GNNs face challenges in preserving commute time due to their requirement for comprehending mutual path relations, which are inherently asymmetric and often involve longer-range interactions especially in digraphs, which are not naturally captured in the basic message-passing framework.

Taking the digraph in Fig. 1 as an example, a 1-layer DirGNN defined in Eq. (1) can encode $v_m$, $v_j$ and $v_k$ into the representation of $v_i$, while also capturing the directionality of edges from these neighbors by using distinct aggregation functions for incoming and outgoing neighbors. It shows that DirGNN can capture the hitting time, as neighbors with lower hitting times, $h(v_i, v_k)$, $h(v_i, v_k)$ and $h(v_m, v_i)$, are aggregated preferentially. However, DirGNN inherently focuses on unidirectional

interactions and overlooks mutual path dependencies. Notably, a 1-layer DirGNN is insufficient for capturing the asymmetric interactions indicated by the paths from $v_j$ and $v_k$ returning to the central node $v_i$ (gray areas). One potential approach to address this limitation is to stack additional message passing layers to encompass the entire commute path between nodes, thereby capturing mutual path interactions. Nevertheless, this strategy is non-trivial because the commute paths vary considerably across different node pairs, complicating the determination of an appropriate number of layers. Additionally, stacking multiple layers to cover these paths can introduce irrelevant non-local information and lead to oversmoothing.

**Goal** We expect to directly encode node-wise commute time into the node representations to accurately reflect the true interaction strength between *adjacent* nodes during neighbor aggregation, accounting for both forward and backward paths. For instance, even though $h(v_i, v_j) = h(v_i, v_k)$, a shorter commute time $c(v_i, v_k) < c(v_i, v_j)$ suggests a stronger interaction from $v_k$ to $v_i$ compared to $v_j$ to $v_i$. Consequently, the contribution of neighbor $v_k$ to the representation of $v_i$ should be greater than that of $v_j$.

### 3.2 COMMUTE TIME COMPUTATION

Based on the standard Markov chain theory, a useful tool to study random walk distances is the fundamental matrix (Aldous & Fill, 2002). We first establish the following assumptions required to support the theorem.

**Assumption 3.1.** *The digraph $G$ is irreducible and aperiodic.*

These two properties pertain to the Markov chain's stationary probability distribution $\pi$ (i.e., Perron vector) corresponding to the given graph. Irreducibility ensures that it is possible to reach any node (state) from any other node, preventing $\pi$ from converging to 0. Aperiodicity ensures that the Markov chain does not get trapped in cycles of a fixed length, thus guaranteeing the existence of a unique $\pi$. Existence and uniqueness of $\pi$ facilitate deterministic analysis and computation. For a more intuitive understanding of the assumptions, we give the sufficient conditions of digraph under the irreducibility and aperiodicity assumptions.

**Proposition 1.** *A strongly connected digraph, in which a directed path exists between every pair of vertices, is irreducible. A digraph with self-loops in each node is aperiodic.*

Given the above assumption, the fundamental matrix $\mathbf{Z}$ is defined as the sum of an infinite matrix series:

$$\mathbf{Z} = \sum_{t=0}^{\infty} \left(\mathbf{P}^t - \mathbf{J}\mathbf{\Pi}\right) = \sum_{t=0}^{\infty} \left(\mathbf{P}^t - e\pi^\top\right), \tag{2}$$

where $e$ is the all-one column vector, then we have $\mathbf{J} = e \cdot e^\top$ is the all-one matrix, and $\mathbf{\Pi} = \mathrm{diag}(\pi)$ is the diagonal matrix of $\pi$.

**Theorem 3.2.** *(Li & Zhang, 2012) Given Assumption 3.1, the fundamental matrix $\mathbf{Z}$ defined in Eq. (2) converges to:*

$$\mathbf{Z} = (\mathbf{I} - \mathbf{P} + \mathbf{J}\mathbf{\Pi})^{-1} - \mathbf{J}\mathbf{\Pi}, \tag{3}$$

*where $\mathbf{I}$ is an identity matrix.*

The hitting time and commute time on $G$ can then be expressed as $\mathbf{Z}$ (Aldous & Fill, 2002) as follows:

$$h(v_i, v_j) = \frac{\mathbf{Z}_{jj} - \mathbf{Z}_{ij}}{\pi_j}, \quad c(v_i, v_j) = h(v_i, v_j) + h(v_j, v_i). \tag{4}$$

However, directly calculating the complete fundamental matrix $\mathbf{Z}$ and the commute times for all node pairs is computationally expensive and yields a dense matrix. Moreover, integrating the random walk distances computation, defined in Eq. (3) and Eq. (4), into the message passing framework is non-trivial, which concerns the scalability of the model.

## 4 Commute Graph Neural Networks

In this section, we present **C**ommute **G**raph **N**eural **N**etworks (CGNN) to encode the commute time information into message passing. We first establish the relationship between random walk distances and the digraph Laplacian.

### 4.1 Digraph Laplacian (`DiLap`)

Contrary to the traditional graph Laplacian, typically defined as a symmetric positive semi-definite matrix derived from the symmetric adjacency matrix, our proposed `DiLap` is built upon the transition matrix to preserve the directed structure. Specifically, the classical graph Laplacian $\mathbf{L} = \mathbf{D} - \mathbf{A}$ is interpreted as the divergence of the gradient of a signal on an undirected graph (Shuman et al., 2013; Hamilton, 2020): given a graph signal $s \in \mathbb{R}^N$, $(\mathbf{L}s)_i = \sum_{j \in \mathcal{N}_i} \mathbf{A}_{ij}(s_i - s_j)$. Intuitively, graph Laplacian corresponds to the difference operator on the signal $s$, and acts as a node-wise measure of local smoothness. In line with this conceptual foundation, we generalize the graph Laplacian to digraphs by defining the divergence of the gradient on digraphs with `DiLap` $\mathbf{T}$:

$$\mathbf{T}s = \mathcal{G}\mathcal{D}s \Longrightarrow \mathbf{T} = \mathbf{B}\mathrm{diag}\left(\{\mathbf{P}_{ij}\}_{(v_i,v_j)\in E}^M\right)\mathbf{B}^\top \tag{5}$$

where $\mathcal{G}$ is the gradient operator on graph signals, and $\mathcal{D}$ is the divergence operator. $\mathbf{B} \in \mathbb{R}^{N\times M}$ is an incidence matrix, where the dimensions represent nodes and edges, respectively. For edge indices $\{e_1, \cdots, e_M\} \in \mathbb{E}$, if $e_k = (v_i, v_j) \in E$, then the $k$-th column of $\mathbf{B}$ corresponding to $e_k$ has $+1$ in row $i$ and $-1$ in row $j$. $\mathrm{diag}\left(\{\mathbf{P}_{ij}\}_{(v_i,v_j)\in E}^M\right)$ is a diagonal matrix whose entries are the transition probabilities corresponding to the edges in the graph. The detailed derivation of $\mathbf{T}$ is included in Appendix A.1, which illustrates how $\mathbf{T}$ functions as a measure of smoothness in directed graphs, taking into account their directional properties. Although the structure of `DiLap` depends on the indices of edges and nodes, such as the ordering of edge transition probabilities in $\mathrm{diag}\left(\{\mathbf{P}_{ij}\}_{(v_i,v_j)\in E}^M\right)$, the following property holds (for proof, see Appendix A.2).

**Proposition 2.** *`DiLap` $\mathbf{T}$ is permutation equivariant with respect to node indices and permutation invariant with respect to edge indices.*

Given the Laplacian operator's role in assessing signal smoothness throughout the graph, it is essential to allocate greater weights to nodes of higher structural importance. This prioritization ensures that the smoothness at nodes central to the graph's structure more significantly influences the overall smoothness measurement. Thus, we further define the Weighted `DiLap` $\mathcal{T}$:

$$(\mathcal{T}s)_i = (\mathbf{\Pi}\mathcal{G}\mathcal{D}s)_i = \pi_i\left(\sum_{v_j \in \mathcal{N}_i^{\mathrm{in}}}(\mathcal{G}s)_{(v_j,v_i)} - \sum_{v_j \in \mathcal{N}_i^{\mathrm{out}}}(\mathcal{G}s)_{(v_i,v_j)}\right) \Longrightarrow \mathcal{T} = \mathbf{\Pi}\mathbf{T} \tag{6}$$

Here we utilize the $i$-th element of the Perron vector $\pi$ to quantify the structural importance of $v_i$, reflecting its eigenvector centrality. This is based on the principle that a node's reachability is directly proportional to its corresponding value in the Perron vector (Xu et al., 2018). Therefore, $\pi$ effectively indicates the centrality and influence over the long term in the graph. Perron-Frobenius Theorem (Horn & Johnson, 2012) establishes that $\pi$ satisfies $\sum_i \pi_i = 1$, is strictly positive, and converges to the left eigenvector of the dominant eigenvalue of $\mathbf{P}$.

### 4.2 Similarity-based Graph Rewiring

Both the fundamental matrix defined in Eq. (3) and Weighted `DiLap` requires Assumption 3.1 to ensure the existence and uniqueness of the Perron vector $\pi$, conditions that are not universally met in general graphs. To fulfill the irreducibility and aperiodicity assumptions, Tong et al. (2020a) introduce a teleporting probability uniformly distributed across all nodes. This method, inspired by PageRank (Page et al., 1999), amends the transition matrix to $\mathbf{P}_{pr} = \gamma\mathbf{P} + (1-\gamma)\frac{ee^\top}{N}$, where $\gamma \in (0,1)$. $\mathbf{P}_{pr}$ allows for the possibility that a random walker may choose a non-neighbor node for

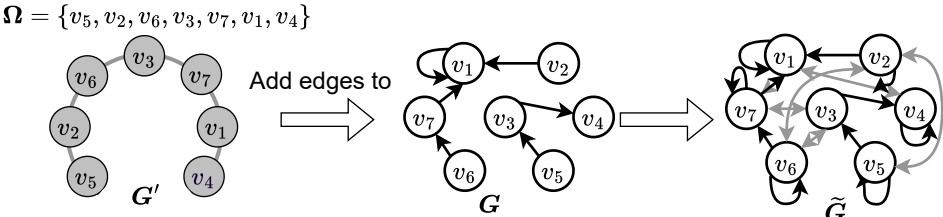

Figure 2: The sorted node indices in $\Omega$ are connected one by one with undirected edges to construct $G'$, then adding all edges from $G'$ to $G$ to generate $\widetilde{G}$.

the next step with a probability of $\frac{1-\gamma}{N}$. This adjustment ensures that $\mathbf{P}_{pr}$ is irreducible and aperiodic, so it has a unique $\pi$. However, this approach leads to a complete graph represented by a dense matrix $\mathbf{P}_{pr}$, posing significant challenges for subsequent computational processes.

Rather than employing $\mathbf{P}_{pr}$ as the transition matrix, we introduce a graph rewiring method based on feature similarity to make a given graph irreducible, while maintaining the sparsity. As outlined in Proposition 1, to transform the digraph into a strongly connected structure, it is essential that each node possesses a directed path to every other node. To this end, we initially construct a simple and irreducible graph $G'$ with all $N$ nodes, then add all edges from $G'$ the original digraph $G$, thereby ensuring $G$'s irreducibility. The construction of $G'$ begins with the calculation of the mean of node features as the anchor vector $\boldsymbol{a}$. Then we determine the similarity between each node and the anchor, sort the similarity values, and return the sorted node indices, denoted as $\Omega \in \mathbb{R}^N$:

$$\boldsymbol{a} = \frac{\sum_i \mathbf{X}_i}{N}, \quad \omega_i = \cos(\boldsymbol{a}, \mathbf{X}_i), \quad S = \arg\text{sort}(\{\omega_i\}_{i=1}^N) \tag{7}$$

where $\cos(\boldsymbol{a}, \mathbf{X}_i)$ is the cosine similarity between node features of $v_i$ and $\boldsymbol{a}$, and $\arg\text{sort}(\cdot)$ yields the indices of nodes that sort similarity values $\{\omega_i\}_{i=1}^N$. We then connect the nodes one by one with undirected (bidirectional) edges following the order in $S$ to construct $G'$, as shown in Fig. 2. Given that $G'$ is strongly connected, adding all its edges into $G$ results in a strongly connected digraph $\widetilde{G}$, which is irreducible. To achieve aperiodicity, self-loops are further added to $\widetilde{G}$.

This rewiring approach satisfies Assumption 3.1 and maintains graph sparsity. Additionally, adding edges between nodes with similar features only minimally alters the overall semantics of the original graph. Based on $\widetilde{G}$ and its corresponding $\widetilde{\mathbf{P}}$, $\widetilde{\mathbf{B}}$, and $\widetilde{\mathbf{\Pi}}$, we have the deterministic Weighted `DiLap` $\widetilde{\mathcal{T}}$.

### 4.3 FROM DILAP TO COMMUTE TIME

Given the Weighted `DiLap` $\widetilde{\mathcal{T}}$, we can unify the commute time information into the message passing by building the connection between $\widetilde{\mathcal{T}}$ and the fundamental matrix $\mathbf{Z}$:

**Lemma 4.1.** *Given a rewired graph $\widetilde{G}$, the Weighted `DiLap` is defined as $\widetilde{\mathcal{T}} = \widetilde{\mathbf{\Pi}}\widetilde{\mathbf{B}}\text{diag}\left(\left\{\widetilde{\mathbf{P}}_{ij}\right\}_{(v_i,v_j)\in E}^M\right)\widetilde{\mathbf{B}}^\top$. Then the fundamental matrix $\mathbf{Z}$ of $\widetilde{G}$ can be solved by:*

$$\mathbf{Z} = \widetilde{\mathcal{T}}^\dagger\widetilde{\mathbf{\Pi}} = \widetilde{\mathbf{T}}^\dagger, \tag{8}$$

*where the superscript $^\dagger$ means Moore–Penrose pseudoinverse of the matrix.*

The proof is given in Appendix A.3. Leveraging Lemma 4.1 and using Eq. (4), we can further compute the hitting times and commute times in terms of $\widetilde{\mathcal{T}}$ with the following theorem.

**Theorem 4.2.** *Given $\widetilde{G}$, the hitting time and commute time from $v_i$ to $v_j$ on $\widetilde{G}$ can be computed as follows:*

$$h(v_i, v_j) = \frac{\widetilde{\mathbf{T}}_{jj}^\dagger}{\pi_j} - \frac{\widetilde{\mathbf{T}}_{ij}^\dagger}{\sqrt{\pi_i\pi_j}},$$

$$c(v_i, v_j) = h(v_i, v_j) + h(v_j, v_i) = \frac{\widetilde{\mathbf{T}}_{jj}^\dagger}{\pi_j} + \frac{\widetilde{\mathbf{T}}_{ii}^\dagger}{\pi_i} - \frac{\widetilde{\mathbf{T}}_{ij}^\dagger}{\sqrt{\pi_i\pi_j}} - \frac{\widetilde{\mathbf{T}}_{ji}^\dagger}{\sqrt{\pi_i\pi_j}}. \tag{9}$$

Then we can derive the matrix forms of the hitting time $\mathcal{H}$ and commute time $\mathcal{C}$ as per Eq. (9):

$$\mathcal{H} = (e \otimes \pi^{-1})(\widetilde{\mathbf{T}}^{\dagger} \odot \mathbf{I}) - \widetilde{\mathbf{T}}^{\dagger} \odot (\pi^{-\frac{1}{2}} \otimes \pi^{-\frac{1}{2}}), \quad \mathcal{C} = \mathcal{H} + \mathcal{H}^{\top} \tag{10}$$

where $\odot$ denotes Hadamard product, and $\otimes$ is outer product. $\mathcal{H}_{ij}$ and $\mathcal{C}_{ij}$ correspond to the hitting and commute time from $v_i$ to $v_j$ respectively. The computation of commute times via `DiLap`, in contrast to the method delineated in Theorem 3.2, is primarily motivated by efficiency concerns. Specifically, Eq. (3) necessitates the inversion of a dense matrix with complexity $\mathcal{O}(N^3)$, whereas our `DiLap`-based method hinges on computing the pseudoinverse of a sparse matrix $\widetilde{\mathbf{T}}$. The pseudoinverse of $\widetilde{\mathbf{T}}$ can be efficiently determined using SVD. Given the sparse nature of $\widetilde{\mathbf{T}}$, we can employ well-established techniques such as the randomized truncated SVD algorithm (Halko et al., 2011; Cai et al., 2023), which takes advantage of sparsity, to reduce the time complexity to $\mathcal{O}(q|E|)$, where $|E|$ denotes the number of edges reflecting the sparsity (See Appendix A.4). Next, we present Commute Graph Neural Networks (CGNN) based on $\mathcal{C}$.

## 4.4 CGNN

$\mathcal{C} \in \mathbb{R}^{N \times N}$ quantifies the strength of mutual relations between node pairs in the random walk context. Notably, smaller values in $\mathcal{C}$ correspond to stronger mutual reachability, indicating stronger relations between node pairs. Thus, $\mathcal{C}$ is a positive symmetric matrix, and the commute-time-based node proximity matrix can be expressed as $\widetilde{\mathcal{C}} = \exp(-\mathcal{C})$. Since the directed adjacency matrix $\mathbf{A}$ represents the outgoing edges of each node, $\mathbf{A}^{\top}$ therefore accounts for all incoming edges. Then we have $\widetilde{\mathcal{C}}^{\text{out}} = \mathbf{A} \odot \widetilde{\mathcal{C}}$ and $\widetilde{\mathcal{C}}^{\text{in}} = \mathbf{A}^{\top} \odot \widetilde{\mathcal{C}}$ represent the proximity between adjacent nodes under outgoing and incoming edges, respectively. We further perform row-wise max-normalization on $\widetilde{\mathcal{C}}^{\text{out}}$ and $\widetilde{\mathcal{C}}^{\text{in}}$ to rescale the maximum value in each row to 1. Given the original graph $G$ as input, we define the $\ell$-th layer of CGNN as:

$$m_{i,\text{in}}^{(\ell)} = \text{Agg}_{\text{in}}^{(\ell)} \left( \left\{ \widetilde{\mathcal{C}}_{ij}^{\text{in}} \cdot h_j^{(\ell-1)} : v_j \in \mathcal{N}_i^{\text{in}} \right\} \right)$$
$$m_{i,\text{out}}^{(\ell)} = \text{Agg}_{\text{out}}^{(\ell)} \left( \left\{ \widetilde{\mathcal{C}}_{ij}^{\text{out}} \cdot h_j^{(\ell-1)} : v_j \in \mathcal{N}_i^{\text{out}} \right\} \right) \tag{11}$$
$$h_i^{(l)} = \text{Comb}^{(\ell)} \left( h_i^{(\ell-1)}, m_{i,\text{in}}^{(\ell)}, m_{i,\text{out}}^{(\ell)} \right),$$

where $\text{Agg}_{\text{in}}^{(\ell)}(\cdot)$ and $\text{Agg}_{\text{out}}^{(\ell)}(\cdot)$ are mean aggregation functions with different feature transformation weights, and $\text{Comb}^{(\ell)}(\cdot)$ is a mean operator. Within each layer, the influence of $v_j$ on the central node $v_i$ is modulated by the commute-time-based proximity $\widetilde{\mathcal{C}}$ based on the edge directionality. We present the pseudocode of CGNN in Algorithm 1.

**Complexity Analysis** The randomized truncated SVD to compute $\widetilde{\mathbf{T}}^{\dagger}$ is $\mathcal{O}(q|E|)$ where $q$ is the required rank, and the message passing iteration has the same time complexity as DirGNN with $\mathcal{O}(L|E|d^2)$. Therefore, the overall time complexity of CGNN is $\mathcal{O}((Ld^2 + q)|E|)$. In practice, $q$ is typically set to 5, rendering the time complexity effectively linear with respect to the number of edges $|E|$. In GNN domain, models with a complexity less than $\mathcal{O}(N^2)$ are generally considered feasible by researchers (Wu et al., 2020). Given that real-world networks are often extremely sparse, i.e., $|E| \ll N^2$, CGNN demonstrates its feasibility as a model within the GNN family.

## 5 EXPERIMENTS

We conduct node classification experiments on eight digraph datasets. Experimental details and data statistics are provided in Appendix C.1 and Appendix C.2. We provide a performance comparison with 12 baselines including 1) General GNNs: GCN (Kipf & Welling, 2017), GAT (Veličković et al., 2018), and GraphSAGE (Hamilton et al., 2017); 2) Non-local GNNs: APPNP (Klicpera et al., 2019), MixHop (Abu-El-Haija et al., 2019), GPRGNN (Chien et al., 2021), and GCNII (Ming Chen et al., 2020); 3) Digraph NNs: DGCN (Tong et al., 2020b), DiGCN (Tong et al., 2020a), MagNet (Zhang et al., 2021), DiGCL (Tong et al., 2021), DUPLEX (Ke et al., 2024), and DirGNN (Rossi et al., 2023). For all baselines, we apply both the symmetrized and asymmetric adjacency matrix for experiments. The results reported are the better of the two results.

Table 1: Node classification results. Accuracy (%) with standard deviation for 10 runs. We high-light/underline the best/second-best method. For general GNN and non-local GNN baselines, we conduct experiments on both symmetrized versions and their directed counterparts, reporting better results from these two settings. OOM indicates out-of-memory. In Table 7 and Table 8 of Appendix D.1, we present detailed experimental results for both directed and undirected settings of all available baselines.

| Method | Squirrel | Chameleon | Citeseer | CoraML | AM-Photo | Snap-Patents | Roman-Empire | Arxiv-Year |
|---|---|---|---|---|---|---|---|---|
| GCN | $52.43_{\pm2.01}$ | $67.96_{\pm1.82}$ | $66.03_{\pm1.88}$ | $70.92_{\pm0.39}$ | $88.52_{\pm0.47}$ | $51.02_{\pm0.06}$ | $73.69_{\pm0.74}$ | $46.02_{\pm0.26}$ |
| GAT | $40.72_{\pm1.55}$ | $60.69_{\pm1.95}$ | $65.58_{\pm1.39}$ | $72.22_{\pm0.57}$ | $88.36_{\pm1.25}$ | OOM | $49.18_{\pm1.35}$ | $45.30_{\pm0.23}$ |
| GraphSAGE | $41.61_{\pm0.74}$ | $62.01_{\pm1.06}$ | $66.81_{\pm1.38}$ | $74.16_{\pm1.55}$ | $89.71_{\pm0.57}$ | $67.45_{\pm0.53}$ | $86.37_{\pm0.80}$ | $55.43_{\pm0.75}$ |
| APPNP | $51.91_{\pm0.56}$ | $45.37_{\pm1.62}$ | $66.90_{\pm1.82}$ | $70.31_{\pm0.67}$ | $87.43_{\pm0.98}$ | $51.23_{\pm0.54}$ | $72.96_{\pm0.38}$ | $50.31_{\pm0.42}$ |
| MixHop | $43.80_{\pm1.48}$ | $60.50_{\pm2.53}$ | $56.09_{\pm2.08}$ | $65.89_{\pm1.50}$ | $87.17_{\pm1.34}$ | $41.22_{\pm0.19}$ | $50.76_{\pm0.14}$ | $45.30_{\pm0.26}$ |
| GPRGNN | $50.56_{\pm1.51}$ | $66.31_{\pm2.05}$ | $61.74_{\pm1.87}$ | $73.31_{\pm1.37}$ | $\underline{90.23_{\pm0.34}}$ | $40.19_{\pm0.03}$ | $64.85_{\pm0.27}$ | $45.07_{\pm0.21}$ |
| GCNII | $38.47_{\pm1.58}$ | $63.86_{\pm3.04}$ | $58.32_{\pm1.93}$ | $64.84_{\pm0.71}$ | $83.40_{\pm0.79}$ | $48.09_{\pm0.09}$ | $74.27_{\pm0.13}$ | $57.36_{\pm0.17}$ |
| DGCN | $37.16_{\pm1.72}$ | $50.77_{\pm3.31}$ | $66.37_{\pm1.93}$ | $75.02_{\pm0.50}$ | $87.74_{\pm1.02}$ | OOM | $51.92_{\pm0.43}$ | OOM |
| DiGCN | $33.44_{\pm2.07}$ | $50.37_{\pm4.31}$ | $64.99_{\pm1.72}$ | $77.03_{\pm0.70}$ | $88.66_{\pm0.51}$ | OOM | $52.71_{\pm0.32}$ | $48.37_{\pm0.19}$ |
| MagNet | $39.01_{\pm1.93}$ | $58.22_{\pm2.87}$ | $65.04_{\pm0.47}$ | $76.32_{\pm0.10}$ | $86.80_{\pm0.65}$ | OOM | $88.07_{\pm0.27}$ | $60.29_{\pm0.27}$ |
| DUPLEX | $57.60_{\pm0.98}$ | $61.25_{\pm0.94}$ | $67.60_{\pm0.72}$ | $72.26_{\pm0.71}$ | $87.80_{\pm0.82}$ | $66.54_{\pm0.11}$ | $79.02_{\pm0.08}$ | $\underline{64.37_{\pm0.27}}$ |
| DiGCL | $35.82_{\pm1.73}$ | $56.45_{\pm2.77}$ | $\underline{67.42_{\pm0.14}}$ | $\mathbf{77.53_{\pm0.14}}$ | $89.41_{\pm0.11}$ | $70.65_{\pm0.07}$ | $87.94_{\pm0.10}$ | $\underline{63.10_{\pm0.06}}$ |
| DirGNN | $75.19_{\pm1.26}$ | $\underline{79.11_{\pm2.28}}$ | $66.57_{\pm0.74}$ | $75.33_{\pm0.32}$ | $88.09_{\pm0.46}$ | $\mathbf{73.95_{\pm0.05}}$ | $\underline{91.23_{\pm0.32}}$ | $64.08_{\pm0.26}$ |
| CGNN | $\mathbf{77.83_{\pm1.52}}$ | $\mathbf{79.62_{\pm2.33}}$ | $\mathbf{71.59_{\pm0.16}}$ | $\underline{77.08_{\pm0.54}}$ | $\mathbf{90.42_{\pm0.10}}$ | $\underline{72.89_{\pm0.24}}$ | $\mathbf{92.87_{\pm0.45}}$ | $\mathbf{66.16_{\pm0.32}}$ |

## 5.1 OVERALL RESULTS AND ANALYSIS

Table 1 reports the node classification results across eight digraph datasets. Our method CGNN achieves new state-of-the-art results on 6 out of 8 datasets, and comparable results on the remaining ones, validating the superiority of CGNN. We provide more observations as follows. Firstly, while non-local GNNs have the potential to cover the commute paths between adjacent nodes by stacking multiple layers, they do not consistently outperform general, shallow GNN models. It suggests that coarsely aggregating all nodes in commute paths is ineffective. The reason is that deeper models may aggregate excessive irrelevant information for the central node. Our goal is to encode mutual relationships between adjacent nodes by considering their commute times. Aggregating all nodes along the entire path introduces excessive information about other nodes unrelated to the direct relationship between the target nodes. Secondly, GNNs tailored for digraphs do not seem to bring substantial gains. Our results show that with careful hyper-parameter tuning, general GNNs can achieve results comparable to, or even better than, some of GNNs tailored for digraphs (DiGCN, MagNet and DiGCL), as evidenced in the Squirrel, Chameleon, and AM-Photo datasets. Thirdly, CGNN achieves state-of-the-art results on both homophilic and heterophilic digraph benchmarks. Notably, DirGNN also performs comparably on heterophilic graphs (e.g., Squirrel and Chameleon), supporting the findings of Rossi et al. (2023) that distinguishing edge directionality during message passing enables the central node to adaptively balance information flows from both heterophilic and homophilic neighbors, effectively mitigating the impact of heterophily. Moreover, CGNN, an enhanced version of DirGNN, further improves performance on these graphs by effectively incorporating commute times to refine the strength of relationships between nodes, enhancing model robustness under heterophily.

To illustrate this, we further examine the relations between commute-time-based proximity and label similarity along edges. As shown in Eq. (11), we use commute-time-based proximity $\widetilde{\mathcal{C}}$ to weigh the neighbors during neighbor aggregation. Then we define a label similarity matrix $\mathcal{M}$ where $\mathcal{M}_{ij} = 1$ if $v_j \in \mathcal{N}_i$ and $y_i = y_j$; otherwise $\mathcal{M}_{ij} = 0$. Essentially, $\mathcal{M}$ extracts the edges connecting nodes with the same classes from the adjacency matrix $\mathbf{A}$. Thus a higher value of $\|\mathcal{M} - (\mathbf{A} + \mathbf{A}^\top)\|_2^2$ indicates a more pronounced negative impact of heterophily on the model's performance. On the other hand, we compute $\|\mathcal{M} - (\widetilde{\mathcal{C}}^{\text{in}} + \widetilde{\mathcal{C}}^{\text{out}})\|_2^2$ to evaluate the efficacy of $\widetilde{\mathcal{C}}$ in filtering heterophilic information. The closer $(\widetilde{\mathcal{C}}^{\text{in}} + \widetilde{\mathcal{C}}^{\text{out}})$ is to $\mathcal{M}$, the more effectively it aids the model in discarding irrelevant heterophilic information. Fig. 3 visually demonstrates these relationships. We observe that in heterophilic datasets, the commute-time-based proximity matrix $(\widetilde{\mathcal{C}}^{\text{in}} + \widetilde{\mathcal{C}}^{\text{out}})$, aligns more closely with the label similarity matrix $\mathcal{M}$ than $(\mathbf{A} + \mathbf{A}^\top)$. It indicates that $\widetilde{\mathcal{C}}$ effectively filters out

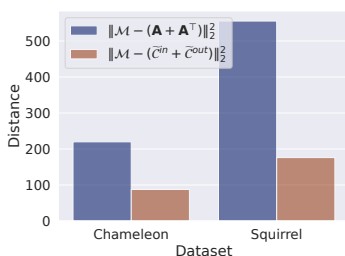

Figure 3: The distance between $\mathcal{M}$ and $\mathbf{A}$, and between $\mathcal{M}$ and $\widetilde{\mathcal{C}}$.

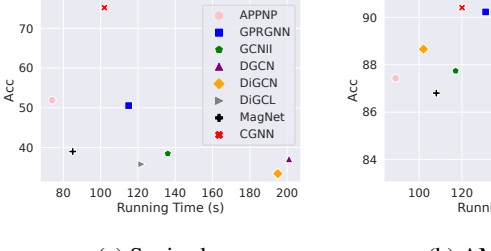

(a) Squirrel     (b) AM-Photo

Figure 4: Accuracy *vs.* running time on Squirrel and AM-Photo.

irrelevant information during message passing by appropriately weighting neighbors, which explains the exceptional performance of CGNN on heterophilic datasets.

**Application scope analysis.** Can commute times *always* enhance message passing on directed graphs? To answer this, we analyze the scope of use for CGNN based on Table 1. For example, on the Snap-Patents and CoraML dataset, we observed that adding commute time-based weights during message passing did not significantly enhance performance. Now we can analyze the reason from the perspective of dataset. CoraML is a directed citation network where nodes predominantly link to other nodes within the same research area. However, in such networks, reciprocal citations between two papers are impossible due to their chronological sequence. Consequently, **mutual path dependencies do not exist**, and thus, incorporating commute times to adjust neighbor weights might might (slightly) hurt performance. A similar situation exists with the Snap-Patents dataset, where each directed edge represents a citation from one patent to another, again indicating the absence of mutual path dependencies.

In conclusion, in networks like citation networks where mutual relationships inherently do not exist, applying commute times to enforce these relationships is unnecessary. Conversely, our model is particularly effective in networks like webpage networks and social networks—examples being Squirrel and AM-Photo—where mutual relationships are prevalent. For instance, in a social network, an ordinary user may follow a celebrity, creating a short path to the celebrity. However, the reverse path from the celebrity back to the user might be considerably longer. Therefore, in such networks, considering mutual relationships based on commute times can provide a more accurate description of node relationships.

## 5.2 Efficiency Comparsion

Fig. 4 compares the accuracy of CGNN and other baseline models with their running times. Despite the additional computational load of calculating commute-time-based proximity, the results show that CGNN provides the best trade-off between effectiveness and efficiency. In particular, on the Squirrel dataset, CGNN has the third-fastest calculation speed while yielding accuracy nearly double that of all other methods. On AM-Photo, CGNN achieves the highest accuracy while maintaining moderate efficiency.

## 5.3 Component Analysis

**Comparison between graph rewiring and PPR.** In Section 4.2, we construct a rewired graph $\widetilde{G}$ based on feature similarity to guarantee the irreducibility and aperiodicity. This approach introduces at most two additional edges per node, specifically targeting those with the highest feature similarity, while minimally altering the original graph structure to preserve semantic information. To in-

Table 2: Changes in commute times before and after rewiring.

|   | CoraML | Chameleon | Roman-Empire |
|---|--------|-----------|--------------|
| $\delta$ | 0.03280 | 0.00157 | 0.06242 |

vestigate changes in commute times before and after rewiring, for the original graph, we use its largest connected component, removing absorbing nodes (i.e., nodes without outgoing edges) to

ensure that we can compute meaningful and deterministic commute times. We denote the average normalized commute time for the original graph as $c_{orig}$; For the rewired graph, we directly compute the commute time and denote this average normalized value as $c_{rew}$. We then use $\delta = \frac{\|c_{orig} - c_{rew}\|_2}{\|c_{orig}\|}$ to quantify the changes, which can be intepretated as the proportion of commute information changed in the original graph. As shown in Table 2, the graph rewiring method can effectively preserve the original commute times of the graph.

In contrast, the classic PageRank transition matrix, defined as $\mathbf{P}_{pr} = \gamma\mathbf{P} + (1 - \gamma)\frac{ee^{\top}}{N}$, achieves a similar objective but results in a completely connected graph $G_{pr}$. However, this approach tends to overlook the sparse structure of the original graph, which may alter the semantic information in the graph. Additionally, computing commute times using a dense transition matrix incurs a high computational cost. To validate the effectiveness of the rewiring approach over the PPR method, we conduct an experiment where $\widetilde{G}$ is replaced with $G_{pr}$ in the computation of commute-time-based proximity. We denote this variant as 'CGNN$_{\text{ppr}}$' and the results of accuracy and efficiency are reported in Table 3. The findings reveal that the PPR approach is suboptimal in terms of both accuracy and efficiency, thereby underscoring the effectiveness of our rewiring-based approach.

Table 3: Accuracy and running time (s) of CGNN and CGNN$_{\text{ppr}}$.

| Method | Squirrel | | Chameleon | | Citeseer | | CoraML | | AM-Photo | |
|---|---|---|---|---|---|---|---|---|---|---|
| | Acc. | Time | Acc. | Time | Acc. | Time | Acc. | Time | Acc. | Time |
| CGNN | **77.83** | **99.25** | **79.62** | **115.77** | **71.59** | **89.25** | **77.08** | **125.20** | **90.42** | **124.17** |
| CGNN$_{\text{ppr}}$ | 68.37 | 257.84 | 71.69 | 253.05 | 68.59 | 137.82 | 76.23 | 192.09 | 88.52 | 203.04 |

**Directed *vs*. Undirected.** To validate the critical role of directed structures in our model, we transform all directed edges into undirected ones by adding their reverse counterparts. This process results in a symmetric adjacency matrix, denoted as $\mathbf{A}_{\text{sym}}$. Subsequently, the commute time is calculated based on the transition matrix derived from

Table 4: Impact of directed structure.

| | Squirrel | CoraML | AM-Photo |
|---|---|---|---|
| CGNN | **77.83** | **77.08** | **90.42** |
| CGNN$_{\text{sym}}$ | 72.37 | 71.29 | 88.53 |

$\mathbf{A}_{\text{sym}}$. We refer this variant as 'CGNN$_{\text{sym}}$'. Table 4 shows the accuracy of CGNN and CGNN$_{\text{sym}}$ on three datasets. We find that edge direction can significantly influence the prediction accuracy for our model.

# 6 LIMITATION

Memory cost represents a limitation of our model. The commute time matrix, $\mathcal{C}$, is inherently dense as it retains commute times between all node pairs, leading to a memory complexity that scales quadratically with the number of nodes, expressed as $\mathcal{O}(N^2)$. In contrast, baseline methods such as GCN, GAT, and DirGNN typically require memory proportional to the number of edges, $\mathcal{O}(M)$s. This distinction highlights the more substantial memory requirements of our approach.

# 7 CONCLUSION

Identifying and encoding asymmetric mutual path dependencies in directed graphs is essential for accurately representing real-world relationships between entities. In this work, we utilize the concept of commute time to assess the strength of relationships in directed graphs and introduce the Commute Graph Neural Network (CGNN) to incorporate node-wise commute time into node representations. To achieve this, we propose `DiLap`, a novel Laplacian formulation derived from the divergence of the gradient of signals on directed graphs, along with an efficient computational method for deterministic commute times. By integrating commute times into GNN message passing through neighbor weighting, CGNN effectively harnesses path asymmetry in directed graphs, thereby enhancing node representation learning. Our extensive experiments across eight directed graph datasets demonstrate that CGNN significantly outperforms existing methods.

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

## A  PROOFS AND DERIVATIONS

### A.1  DERIVATION OF `DiLap` T

The gradient operator $\mathcal{G}$ maps a signal defined on the nodes of the graph to a signal on the edges. For a directed graph $G$ and a signal $s \in \mathbb{R}^N$ on the nodes, the gradient $\mathcal{G}s$ is defined on the edges as:

$$(\mathcal{G}s)_{(v_i, v_j)} = \mathbf{P}_{ij}(s_i - s_j) \tag{12}$$

for each directed edge $(v_i, v_j) \in E$. This captures the difference in the signal between the source node $v_i$ and the target node $v_j$.

The divergence operator $\mathcal{D}$ maps a signal defined on the edges back to a signal on the nodes. For a signal $\mathcal{G}s \in \mathbb{R}^M$ on the edges, the divergence at node $v_i$ is:

$$(\mathcal{D}(\mathcal{G}s))_i = \sum_{v_j \in \mathcal{N}_i^{\text{in}}} (\mathcal{G}s)_{(v_j, v_i)} - \sum_{v_j \in \mathcal{N}_i^{\text{out}}} (\mathcal{G}s)_{(v_i, v_j)} \tag{13}$$

This computes the net "incoming" minus "outgoing" signal flow at each node. The digraph Laplacian `DiLap` T is then defined as the composition of the divergence and gradient operators on the original signal $s$:

$$\mathbf{T}s = \mathcal{D}\mathcal{G}s \tag{14}$$

Eq. (12) and Eq. (13) demonstrate that the composed operator forming `DiLap` effectively measures how the signal diverges from each node considering the graph's directionality. Therefore, analogous to the traditional Laplacian in undirected graphs, `DiLap` acts as a measure of smoothness specifically tailored for directed graphs.

To express $\mathbf{T}$ in matrix form, we initially define the incidence matrix $\mathbf{B} \in \mathbb{R}^{N \times M}$, which encapsulates both the connectivity and the directionality of the edges within the digraph:

$$\mathbf{B}_{ik} = \begin{cases} +1, & e_k = (v_i, v_j) \\ -1, & e_k = (v_j, v_i) \,, \\ 0, & \text{otherwise} \end{cases} \tag{15}$$

where $k \in \{1, \cdots, M\}$ represents fixed edge indices, and each undirected edge is treated as comprising two **uni**directional edges. Then We construct a diagonal matrix representing the edge transition probabilities, denoted as $\mathrm{diag}\left(\{\mathbf{P}_{ij}\}_{(v_i, v_j) \in E}^M\right) \in \mathbb{R}^{M \times M}$, where the principal diagonal elements are indexed according to the edge indices. Based on the above definitions, the gradient operator $\mathcal{G}$ can be represented as $\mathcal{G} = \mathrm{diag}\left(\{\mathbf{P}_{ij}\}_{(v_i, v_j) \in E}^M\right) \mathbf{B}^\top$, and the divergence operator as $\mathcal{D} = \mathbf{B}$. Therefore, the `DiLap` becomes:

$$\mathbf{T} = \mathbf{B}\mathrm{diag}\left(\{\mathbf{P}_{ij}\}_{(v_i, v_j) \in E}^M\right) \mathbf{B}^\top \tag{16}$$

## A.2 PROOF OF PROPOSITION 2

*Proof.* Let $\mathbf{Q}_{\mathrm{node}} \in \mathbb{R}^{N \times N}$ be a node permutation matrix that reorders the nodes in $G$. The permuted incidence matrix can be represented as $\mathbf{B}' = \mathbf{Q}_{\mathrm{node}}^\top \mathbf{B}$. Then we have the permuted `DiLap` $\mathbf{T}'$:

$$\begin{aligned} \mathbf{T}' &= \mathbf{B}'\mathrm{diag}\left(\{\mathbf{P}_{ij}\}_{(v_i, v_j) \in E}^M\right) \mathbf{B}'^\top \\ &= \left(\mathbf{Q}_{\mathrm{node}}^\top \mathbf{B}\right) \mathrm{diag}\left(\{\mathbf{P}_{ij}\}_{(v_i, v_j) \in E}^M\right) \left(\mathbf{B}^\top \mathbf{Q}_{\mathrm{node}}\right) \\ &= \mathbf{Q}_{\mathrm{node}}^\top \mathbf{T} \mathbf{Q}_{\mathrm{node}} \end{aligned} \tag{17}$$

Eq. (17) shows that $\mathbf{T}'$ is obtained by conjugating $\mathbf{T}$ with the node permutation matrix $\mathbf{Q}_{\mathrm{node}}$, which means $\mathbf{T}'$ is $\mathbf{T}$ with its rows and columns permuted according to $\mathbf{Q}_{\mathrm{node}}$. Thus $\mathbf{T}$ is permutation equivariant up to a relabeling of nodes.

Let $\mathbf{Q}_{\mathrm{edge}} \in \mathbb{R}^{M \times M}$ be an edge permutation matrix that reorders the edges of $G$. The permuted incidence matrix can be represented as $\mathbf{B}' = \mathbf{B}\mathbf{Q}_{\mathrm{edge}}$, and the permuted diagonal matrix $\mathrm{diag}\left(\{\mathbf{P}_{ij}\}_{(v_i, v_j) \in E}^M\right)' = \mathbf{Q}_{\mathrm{edge}}^\top \mathrm{diag}\left(\{\mathbf{P}_{ij}\}_{(v_i, v_j) \in E}^M\right) \mathbf{Q}_{\mathrm{edge}}$. Then we have the permuted `DiLap` $\mathbf{T}'$:

$$\begin{aligned} \mathbf{T}' &= \mathbf{B}'\mathrm{diag}\left(\{\mathbf{P}_{ij}\}_{(v_i, v_j) \in E}^M\right)' \mathbf{B}'^\top \\ &= \left(\mathbf{B}\mathbf{Q}_{\mathrm{edge}}\right) \left(\mathbf{Q}_{\mathrm{edge}}^\top \mathrm{diag}\left(\{\mathbf{P}_{ij}\}_{(v_i, v_j) \in E}^M\right) \mathbf{Q}_{\mathrm{edge}}\right) \left(\mathbf{Q}_{\mathrm{edge}}^\top \mathbf{B}^\top\right) \\ &= \mathbf{B}\mathrm{diag}\left(\{\mathbf{P}_{ij}\}_{(v_i, v_j) \in E}^M\right) \mathbf{B}^\top \\ &= \mathbf{T} \end{aligned} \tag{18}$$

Eq. (18) shows that $\mathbf{T}$ remains unchanged under edge permutation when $\mathbf{B}$ and $\mathrm{diag}\left(\{\mathbf{P}_{ij}\}_{(v_i, v_j) \in E}^M\right)$ are adjusted accordingly. Thus $\mathbf{T}$ is fully invariant to the ordering of edges. $\square$

## A.3 PROOF OF LEMMA 4.1

*Proof.* We first define the weighted out-transition matrix as $\mathbf{F} = \mathrm{diag}\left(\left\{\pi_i \sum_j \mathbf{P}_{ij}\right\}_{i=1}^N\right)$. Based on $\mathbf{F}$, the weight `DiLap` $\mathcal{T}$ can be written as $\mathcal{T} = \mathbf{F} - \mathbf{\Pi}\mathbf{P}$. $\mathbf{P}$ can be expressed as:

$$\mathbf{P} = \mathbf{\Pi}^{-1}(\mathbf{F} - \mathcal{T}). \tag{19}$$

Since the transition matrix $\mathbf{P}$ is row-stochastic, it follows that $\mathbf{P}^t \mathbf{J} = \mathbf{J}$. In light of Eq. (2) and considering that $\pi$ is stochastic, we have $\mathbf{Z}\mathbf{J} = \mathbf{0}_{n \times n}$ and $\mathbf{\Pi}^{-\frac{1}{2}} \mathbf{F} \mathbf{\Pi}^{-\frac{1}{2}} = \mathbf{I}$.

Let $\mathcal{K} = \mathbf{\Pi}^{-\frac{1}{2}} \mathcal{T} \mathbf{\Pi}^{-\frac{1}{2}}$, $\mathcal{J} = \mathbf{\Pi}^{\frac{1}{2}} \mathbf{J} \mathbf{\Pi}^{\frac{1}{2}}$, and $\mathcal{Z} = \mathbf{\Pi}^{\frac{1}{2}} \mathbf{Z} \mathbf{\Pi}^{-\frac{1}{2}}$, we have $\mathcal{J}^2 = \mathcal{J}$. As $\pi^\top \mathbf{Z} = \mathbf{0}$, we have $\mathcal{Z}\mathcal{J} = \mathbf{\Pi}^{\frac{1}{2}} \mathbf{Z} \mathbf{J} \mathbf{\Pi}^{\frac{1}{2}} = \mathbf{0}_{N \times N}$ and $\mathcal{J}\mathcal{Z} = \mathbf{0}_{N \times N}$. Since $\mathbf{B}^\top \mathbf{J} = \mathbf{0}_{N \times N}$, $\mathbf{T}\mathbf{J} = \mathbf{0}_{N \times N}$ and $\mathbf{J}\mathbf{T} = \mathbf{0}_{N \times N}$ holds. Incorporating these into Eq. (3), we have:

$$\mathcal{Z} + \mathcal{J} = (\mathcal{K} + \mathcal{J})^{-1}. \tag{20}$$

By post-multiplying Eq. (20) from the right by $(\mathcal{K} + \mathcal{J})$, we have:

$$\mathbf{I} - \mathcal{J} = \mathcal{Z}\mathcal{K} + \mathcal{J}\mathcal{K}, \tag{21}$$

where $\mathcal{J}\mathcal{K} = (\mathbf{\Pi}^{\frac{1}{2}}\mathbf{J}\mathbf{\Pi}^{\frac{1}{2}})(\mathbf{\Pi}^{-\frac{1}{2}}\mathbf{T}\mathbf{\Pi}\mathbf{\Pi}^{-\frac{1}{2}}) = \mathbf{\Pi}^{\frac{1}{2}}\mathbf{J}\mathbf{T}\mathbf{\Pi}^{-\frac{1}{2}} = \mathbf{0}_{N \times N}$. Then we have:

$$\mathcal{Z}\mathcal{K} = \mathbf{I} - \mathcal{J} \tag{22}$$

Similarly, by multiplying from the left, we establish that $\mathcal{K}\mathcal{Z} = \mathbf{I} - \mathcal{J}$. Since $\mathcal{J}\mathcal{Z} = \mathbf{0}_{N \times N}$, $\mathcal{Z}\mathcal{K}\mathcal{Z} = \mathcal{Z}$. Furthermore, $\mathcal{K}\mathcal{J} = (\mathbf{\Pi}^{-\frac{1}{2}}\mathbf{T}\mathbf{\Pi}\mathbf{\Pi}^{-\frac{1}{2}})(\mathbf{\Pi}^{\frac{1}{2}}\mathbf{J}\mathbf{\Pi}^{\frac{1}{2}}) = \mathbf{0}_{N \times N}$ leads to $\mathcal{K}\mathcal{Z}\mathcal{K} = \mathcal{K}$. Considering the symmetry of the left part of Eq. (23), we have $(\mathcal{Z}\mathcal{K})^{\top} = \mathcal{Z}\mathcal{K}$. Similarly, $(\mathcal{K}\mathcal{Z})^{\top} = \mathcal{K}\mathcal{Z}$. These derivations satisfy the sufficient conditions for the Moore–Penrose pseudoinverse, such that

$$\mathcal{Z} = \mathcal{K}^{\dagger} \tag{23}$$

Finally, recovering $\mathcal{Z}$ and $\mathcal{K}$ as:

$$\mathbf{Z} = \mathcal{T}^{\dagger}\mathbf{\Pi} \tag{24}$$

which concludes the proof. $\qquad\square$

### A.4 SVD FOR $\widetilde{\mathbf{T}}^{\dagger}$

Given a matrix $\widetilde{\mathbf{T}} \in \mathbb{R}^{N \times N}$, its Moore-Penrose pseudoinverse can be directly computed with an SVD-based method. Specifically, we first perform truncated SVD on $\widetilde{\mathbf{T}} \approx \mathbf{U}_q \Sigma_q \mathbf{V}_q^{\top}$, where $\mathbf{U}_q \in \mathbb{R}^{N \times q}$ and $\mathbf{V}_q \in \mathbb{R}^{N \times q}$ contains the first $q$ columns of $\mathbf{U}$ and $\mathbf{V}$. $\Sigma_q \in \mathbb{R}^{q \times q}$ is the diagonal matrix of $q$ largest singular values. It is a $q$-rank approximation of $\widetilde{\mathbf{T}}$, which holds that $\mathrm{rank}(\mathrm{R}) = q$. Then the Moore-Penrose pseudoinverse of $\widetilde{\mathbf{T}}$ can be easily computed as follows:

$$\widetilde{\mathbf{T}}^{\dagger} = \mathbf{U}_q \Sigma_q^{-1} \mathbf{V}_q^{\top}. \tag{25}$$

To leverage sparsity of $\widetilde{\mathbf{T}}$ to avoid $\mathcal{O}(N^3)$ complexity, we adopt the randomized SVD algorithm proposed by (Halko et al., 2011; Cai et al., 2023) to first approximate the range of the input matrix with a low-rank orthonormal matrix, and then perform SVD on this smaller matrix:

$$\hat{\boldsymbol{U}}_q, \hat{\Sigma}_q, \hat{\boldsymbol{V}}_q^{\top} = \mathrm{ApproxSVD}(\widetilde{\mathbf{T}}, q), \quad \hat{\widetilde{\mathbf{T}}}_{SVD} = \hat{\boldsymbol{U}}_q \hat{\Sigma}_q \hat{\boldsymbol{V}}_q^{\top}, \tag{26}$$

where $\hat{\boldsymbol{U}}_q$, $\hat{\Sigma}_q$, and $\hat{\boldsymbol{V}}_q$ are the approximated versions of $\mathbf{U}_q$, $\Sigma_q$, and $\mathbf{V}_q$. Then the Moore-Penrose pseudoinverse of $\widetilde{\mathbf{T}}$ can be computed by:

$$\widetilde{\mathbf{T}}^{\dagger} = \hat{\boldsymbol{U}}_q \hat{\Sigma}_q^{-1} \hat{\boldsymbol{V}}_q^{\top}. \tag{27}$$

The computation cost of randomized truncated SVD takes $\mathcal{O}(qK)$, where $K$ is the number of non-zero elements in $\widetilde{\mathbf{T}}$, so we have $K = |E|$. Thus, the sparsity degree of $\widetilde{\mathbf{T}}$ can determine the time complexity of its Moore-Penrose pseudoinverse, which demonstrates the importance of Lemma 4.1.

## B PSEUDO CODE FOR CGNN

## C IMPLEMENTATION DETAILS

### C.1 EXPERIMENTAL SETTINGS

We evaluate the performance by node classification accuracy with standard deviation in the semi-supervised setting. For Squirrel and Chameleon, we use 10 public splits (48%/32%/20% for training/validation/testing) provided by (Pei et al., 2019). For the remaining datasets, we adopt the same splits as (Tong et al., 2020a; 2021), which chooses 20 nodes per class for the training set, 500 for the validation set, and allocates the rest to the test set. We conduct our experiments on 2 Intel Xeon Gold 5215 CPUs and 1 NVIDIA GeForce RTX 3090 GPU.

### C.2 DATA STATISTICS

The datasets used in Section 5 are Squirrel, Chameleon (Rozemberczki et al., 2021), Citeseer (Sen et al., 2008), CoraML (Bojchevski & Günnemann, 2017), AM-Photo (Shchur et al., 2018), Snap-Patents, Roman-Empire, and Arxiv-Year (Rossi et al., 2023). We summarize their statistics in Table 5. homo_ratio represents the homophily ratio, a metric proposed by Zhu et al. (2020). which is employed to gauge the degree of homophily within the graph. A lower homo_ratio signifies a greater degree of heterophily, indicating a higher prevalence of edges that connect nodes of differing classes.

---

**Algorithm 1** CGNN

---

**Input:** Digraph $G = (V, E, \mathbf{X})$; Depth $L$; Hidden size $d'$; Number of classes $K$
**Output:** Logits $\hat{Y} \in \mathbb{R}^{N \times K}$

1: Compute the anchor $\boldsymbol{a}$ and node-anchor similarities to construct $G'$ with Eq. (7).
2: Add all edges from $G'$ to $G$ to generate $\widetilde{G}$.
3: Compute the Weight `DiLap` $\widetilde{\mathcal{T}}$ for $\widetilde{G}$ with Eq. (6).
4: Compute $\mathcal{R}$ and its Moore-Penrose pseudoinverse with Eq. (8) and Eq. (27).
5: Compute the commute time matrix $\mathcal{C}$ with Eq. (10).
6: Compute the normalized proximity matrix $\widetilde{\mathcal{C}}$ with $\widetilde{\mathcal{C}}^{\text{out}} = \mathbf{A} \odot \widetilde{\mathcal{C}}$ and $\widetilde{\mathcal{C}}^{\text{in}} = \mathbf{A}^\top \odot \widetilde{\mathcal{C}}$.
7: **for** $\ell \in \{1, \cdots, L\}$ **do**
8:     Layer-wise message passing with Eq. (11).
9: **end for**
10: $\mathbf{H} = \text{MLP}(\mathbf{H}^{(L)})$.
11: $\hat{Y} = \text{Softmax}(\mathbf{H})$.

---

Table 5: Statistics of the datasets.

| Dataset | $N$ | $|E|$ | # Feat. | # Classes | homo_ratio |
|---|---|---|---|---|---|
| Squirrel | 5,201 | 217,073 | 2,089 | 5 | 0.22 |
| Chameleon | 2,277 | 36,101 | 2,325 | 5 | 0.23 |
| Cora-ML | 2,995 | 8,416 | 2,879 | 7 | 0.79 |
| Citeseer | 3,312 | 4,715 | 3,703 | 6 | 0.74 |
| AM-Photo | 7,650 | 238,162 | 745 | 8 | 0.83 |
| Snap-Patents | 2,923,922 | 13,975,791 | 269 | 5 | 0.22 |
| Roman-Empire | 22,662 | 44,363 | 300 | 18 | 0.05 |
| Arxiv-Year | 169,343 | 1,166,243 | 128 | 40 | 0.22 |

### C.3 HYPERPARAMETER SETTINGS

For our model, we tune the hyperparameters based on the highest average validation accuracy. We utilize the randomized truncated SVD algorithm for computing the Moore-Penrose pseudoinverse of matrix $\mathcal{R}$, setting the required rank $q$ to 5 for all datasets. The learning rate $lr$ is selected from $\{0.01, 0.005\}$, and the weight decay $wd$ from $\{0, 5e-5, 5e-4\}$. In the model architecture, the number of layers $L$ vary among $\{1, 2, 3, 4, 5\}$ and the dimension $d'$ is selected from $\{32, 64, 128, 256, 512\}$. The comprehensive hyperparameter configurations for CGNN are detailed in Table 6.

Table 6: Hyperparameters specifications.

| Dataset | $lr$ | $wd$ | $L$ | $d'$ |
|---|---|---|---|---|
| Squirrel | 0.005 | 0 | 5 | 128 |
| Chameleon | 0.01 | 0 | 4 | 128 |
| CoraML | 0.01 | 0 | 2 | 64 |
| Citeseer | 0.01 | 0 | 2 | 128 |
| AM-Photo | 0.005 | 0 | 2 | 512 |
| Snap-Patents | 0.01 | 0 | 2 | 32 |
| Roman-Empire | 0.01 | $5e-4$ | 2 | 64 |
| Arxiv-Year | 0.01 | $5e-4$ | 2 | 64 |

## D ADDITIONAL EXPERIMENTS

### D.1 DETAILED EXPERIMENTAL RESULTS ON NODE CLASSIFICATION

Table 1 in Section 5 presents the results from experiments conducted on all eight directed graph datasets. For each baseline, experiments were carried out on both the original directed graph datasets

and their undirected counterparts, which feature symmetrized adjacency matrices. The superior accuracy results from these two settings are reported in Table 1. This section provides a detailed exposition of the experimental outcomes for these configurations in Table 7 and Table 8. It is important to note that while GCN is traditionally a spectral method suited only for undirected graphs, it can be adapted to directed graphs by interpreting it from a spatial perspective, specifically, by aggregating outgoing neighbors with the weight $\frac{1}{\sqrt{d_i d_j}}$. This adaptation allows GCN to be applicable in both experimental settings. Additionally, APPNP, GPRGNN, and GCNII are spectral methods that require symmetrized adjacency matrices for spectral filters. Therefore, we only report their results under the undirected settings in Table 1. For DirGNN and CGNN, in the case of undirected graphs, these models degenerate to GraphSAGE.

Table 7: Comparison of node classification accuracy between original directed graphs and their undirected counterparts on Squirrel, Chameleon, Citeseer, and CoraML.

| Method | Squirrel | | Chameleon | | Citeseer | | CoraML | |
| | Dir. | Undir. | Dir. | Undir. | Dir. | Undir. | Dir. | Undir. |
|---|---|---|---|---|---|---|---|---|
| GCN | $52.43_{\pm 2.01}$ | $51.93_{\pm 1.19}$ | $63.37_{\pm 0.92}$ | $67.96_{\pm 1.82}$ | $64.27_{\pm 1.56}$ | $66.03_{\pm 1.88}$ | $68.73_{\pm 0.24}$ | $70.92_{\pm 0.39}$ |
| GAT | $40.72_{\pm 1.55}$ | $40.50_{\pm 1.47}$ | $60.69_{\pm 1.95}$ | $59.37_{\pm 1.52}$ | $65.58_{\pm 1.39}$ | $54.22_{\pm 0.98}$ | $72.20_{\pm 0.49}$ | $72.22_{\pm 0.57}$ |
| GraphSAGE | $35.19_{\pm 0.54}$ | $41.61_{\pm 0.74}$ | $58.20_{\pm 1.19}$ | $62.01_{\pm 1.06}$ | $62.57_{\pm 0.71}$ | $66.81_{\pm 1.38}$ | $74.16_{\pm 1.55}$ | $72.98_{\pm 0.90}$ |
| MixHop | $39.25_{\pm 0.91}$ | $43.80_{\pm 1.48}$ | $60.50_{\pm 2.53}$ | $60.15_{\pm 1.22}$ | $56.09_{\pm 2.08}$ | $54.71_{\pm 0.50}$ | $65.89_{\pm 1.50}$ | $61.20_{\pm 0.91}$ |
| DGCN | $37.16_{\pm 1.72}$ | $38.24_{\pm 1.19}$ | $50.7_{\pm 3.31}$ | $48.26_{\pm 1.97}$ | $66.37_{\pm 1.93}$ | $62.15_{\pm 0.80}$ | $75.02_{\pm 0.50}$ | $73.11_{\pm 0.68}$ |
| DiGCN | $33.44_{\pm 2.07}$ | $28.17_{\pm 1.90}$ | $50.37_{\pm 4.31}$ | $43.08_{\pm 5.77}$ | $64.99_{\pm 1.72}$ | $64.35_{\pm 1.64}$ | $77.03_{\pm 0.70}$ | $76.98_{\pm 1.00}$ |
| MagNet | $39.01_{\pm 1.93}$ | $35.20_{\pm 1.65}$ | $58.22_{\pm 2.87}$ | $55.46_{\pm 3.10}$ | $65.04_{\pm 0.47}$ | $64.90_{\pm 0.51}$ | $76.32_{\pm 0.10}$ | $76.29_{\pm 0.08}$ |
| DUPLEX | $57.60_{\pm 0.98}$ | $55.26_{\pm 1.10}$ | $61.25_{\pm 0.94}$ | $61.20_{\pm 0.75}$ | $67.60_{\pm 0.72}$ | $67.35_{\pm 0.70}$ | $72.26_{\pm 0.71}$ | $72.21_{\pm 0.65}$ |
| DiGCL | $35.82_{\pm 1.73}$ | $33.10_{\pm 0.94}$ | $56.45_{\pm 2.77}$ | $51.16_{\pm 3.85}$ | $67.42_{\pm 0.14}$ | $66.53_{\pm 0.10}$ | $77.53_{\pm 0.14}$ | $76.24_{\pm 0.05}$ |

Table 8: Comparison of node classification accuracy between original directed graphs and their undirected counterparts on AM-Photo, Snap-Patents, Roman-Empire, and Arxiv-Year.

| Method | AM-Photo | | Snap-Patents | | Roman-Empire | | Arxiv-Year | |
| | Dir. | Undir. | Dir. | Undir. | Dir. | Undir. | Dir. | Undir. |
|---|---|---|---|---|---|---|---|---|
| GCN | $88.52_{\pm 0.47}$ | $85.33_{\pm 0.25}$ | $51.02_{\pm 0.06}$ | $50.15_{\pm 0.04}$ | $73.69_{\pm 0.74}$ | $73.58_{\pm 0.37}$ | $46.02_{\pm 0.26}$ | $44.81_{\pm 0.19}$ |
| GAT | $88.36_{\pm 1.25}$ | $87.50_{\pm 1.77}$ | OOM | OOM | $49.18_{\pm 1.35}$ | $43.37_{\pm 1.02}$ | $45.30_{\pm 0.23}$ | $43.27_{\pm 0.09}$ |
| GraphSAGE | $89.71_{\pm 0.57}$ | $86.23_{\pm 1.25}$ | $67.45_{\pm 0.53}$ | $60.10_{\pm 0.26}$ | $86.37_{\pm 0.80}$ | $84.26_{\pm 0.28}$ | $55.43_{\pm 0.75}$ | $51.19_{\pm 0.73}$ |
| MixHop | $87.17_{\pm 1.30}$ | $85.50_{\pm 1.01}$ | $40.17_{\pm 0.10}$ | $41.22_{\pm 0.19}$ | $43.00_{\pm 0.06}$ | $50.76_{\pm 0.14}$ | $45.30_{\pm 0.26}$ | $41.25_{\pm 0.50}$ |
| DGCN | $87.74_{\pm 1.02}$ | $86.53_{\pm 1.77}$ | OOM | OOM | $51.92_{\pm 0.43}$ | $50.50_{\pm 0.47}$ | OOM | OOM |
| DiGCN | $88.66_{\pm 0.51}$ | $87.94_{\pm 0.23}$ | OOM | OOM | $52.71_{\pm 0.32}$ | $50.43_{\pm 0.21}$ | $48.37_{\pm 0.19}$ | $47.26_{\pm 0.11}$ |
| MagNet | $86.80_{\pm 0.65}$ | $85.21_{\pm 0.20}$ | OOM | OOM | $88.07_{\pm 0.27}$ | $82.99_{\pm 0.80}$ | $60.29_{\pm 0.27}$ | $55.25_{\pm 0.10}$ |
| DUPLEX | $85.19_{\pm 0.73}$ | $87.80_{\pm 0.82}$ | $64.92_{\pm 0.10}$ | $66.54_{\pm 0.11}$ | $79.02_{\pm 0.08}$ | $77.64_{\pm 0.07}$ | $64.37_{\pm 0.27}$ | $62.12_{\pm 0.18}$ |
| DiGCL | $89.41_{\pm 0.11}$ | $87.36_{\pm 0.20}$ | $70.65_{\pm 0.07}$ | $68.62_{\pm 0.08}$ | $87.94_{\pm 0.10}$ | $84.00_{\pm 0.28}$ | $63.10_{\pm 0.06}$ | $59.02_{\pm 0.02}$ |

## D.2 SENSITIVITY ANALYSIS

We investigate the sensitivity of CGNN to key hyperparameters that influence its performance, specifically focusing on the number of layers $L$ and the dimension of the hidden layer $d'$. We explore a range of values for $L$, considering $\{1, 2, 3, 4, 5\}$, and for $d'$, considering $\{32, 64, 128, 256, 512\}$. From Fig. 5, we observe that we observe that CGNN achieves optimal performance with $L = 5$ and $d' = 128$ on Squirrel, and with $L = 2$ and $d' = 64$ on CoraML. This suggests that deeper models are necessary to effectively aggregate valuable information in heterophilic graphs, whereas in homophilic graphs, leveraging local neighborhood information is generally adequate.

## D.3 LABEL SIMILARITY

Recall from Section 5.1, where we investigate the effectiveness of commute time in enhancing GNN performance. We compare the squared Frobenius norm of the differences between the label similarity matrix, $\mathcal{M}$, and two propagation matrices: the commute-time-based propagation matrix $\tilde{\mathcal{C}}^{\text{in}} + \tilde{\mathcal{C}}^{\text{out}}$, and the original propagation matrix $\mathbf{A} + \mathbf{A}^\top$. This comparison aims to assess how well commute time facilitates the discarding of irrelevant heterophilic information during message passing. In

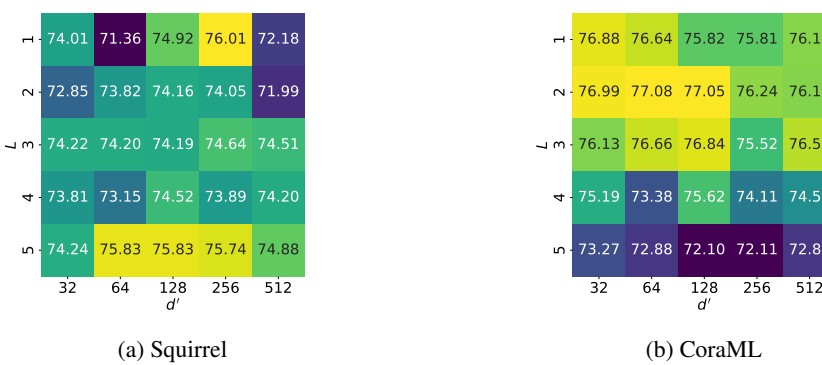

(a) Squirrel        (b) CoraML

Figure 5: Sensitivity analysis on Squirrel and CoraML.

Fig. 6a, we analyze the label similarity in homophilic graphs, specifically CoraML and Citeseer. The results demonstrate that our model effectively filters and enhances useful information even within homophilic graph settings. Additionally, we consider two other propagation matrices: the matrix $\widehat{\widehat{\mathbf{A}}}$ from vanilla GCN used in vanilla GCN, and the approximate personalized PageRank APPR from PPRGo (Bojchevski et al., 2020), further broadening our comparative framework. Fig. 6b and and Fig. 6d illustrate that GCN and PPRGo only slightly reduce heterophilic information from neighbors during message passing, in contrast to the more substantial reductions achieved by CGNN.

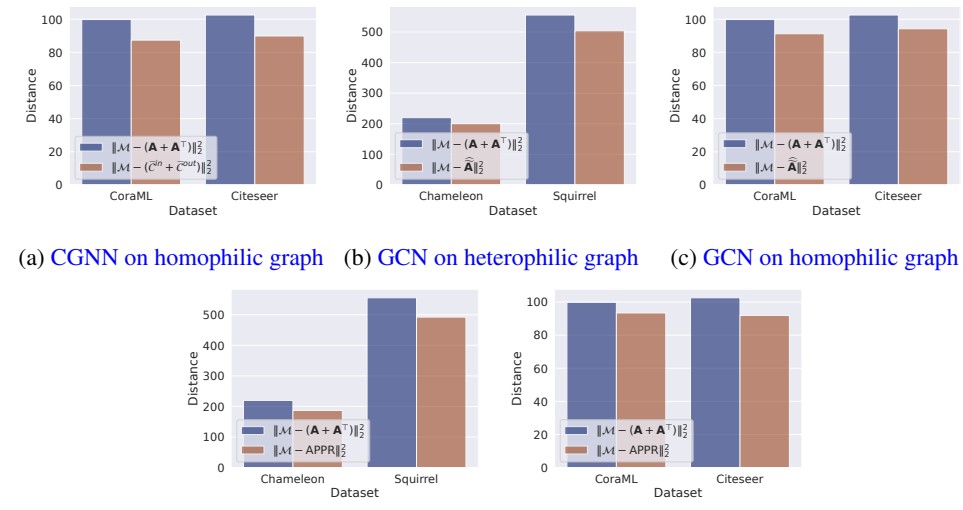

(a) CGNN on homophilic graph    (b) GCN on heterophilic graph    (c) GCN on homophilic graph

(d) PPRGo on heterophilic graph    (e) PPRGo on heterophilic graph

Figure 6: The distance between $\mathcal{M}$ and $\mathbf{A}$, as well as between $\mathcal{M}$ and the propagation matrices used by CGNN, GCN, and PPRGo on both heterophilic and homophilic graphs.

### D.4 SYNTHETIC DATASET

To intuitively examine how commute time enhances the GNN's ability to learn node relationships, we introduce a synthetic dataset tailored for binary classification on directed graphs. This dataset comprises 3,000 nodes, evenly split into two classes of 1,500 nodes each. Node features for each class are generated from distinct Gaussian distributions: $\mathcal{N}(0, 1)$ for the first class and $\mathcal{N}(3, 1)$ for the second.

Edge construction within this dataset adheres to class-based probabilities: nodes within the same class connect with a probability of 0.2, whereas inter-class connections occur at a probability of 0.02, with all connections assigned random directions. Additionally, we define a commute path length

range between [2, 7], creating a graph where each node has an asymmetric commute path with its neighbors. This method allows us to create a graph where each node has an asymmetric commute path with its neighbors, facilitating a detailed examination of how graph neural networks perform under varying structural conditions.

Upon this graph, we deploy our CGNN model to learn node representations. We then measure the Mutual Information (MI) between the central node and its neighbors, differentiated by short ($\alpha_s$) and long ($\alpha_l$) commute times. A higher average MI for short commute times ($\overline{\alpha}_s$) compared to long commute times ($\overline{\alpha}_l$) would validate our model's capacity to effectively capture and preserve commute relationships. Our empirical results reveal that CGNN attained an $\overline{\alpha}_s = 13.2974$ and an $\overline{\alpha}_l = 6.5521$, corroborating the intended design and efficacy of our model in leveraging commute times for enhanced node representation learning.

### D.5 IMPACT OF GRAPH REWIRING

To explore how the rewiring procedure only minimally alters the overall semantics of the original graph, we define edge density as $\delta = \frac{M}{M_{\max}}$, where $M_{\max}$ is the maximum possible number of edges ($N^2$ for both $G$ and $\widetilde{G}$) in the graph and $M$ is the actual number of edges. We denote the edge density of the original graph $G$ as $\delta$ and that of

Table 9: Impact of directed structure.

|  | AM-Photo | Snap-Patent | Arxiv-Year |
|---|---|---|---|
| $\Delta$ | 0.103 | 0.067 | 0.032 |

the rewired graph $\widetilde{G}$ as $\widetilde{\delta}$. Thus the change of graph density after rewiring can be represented as $\Delta = \frac{\widetilde{\delta}-\delta}{\delta} \in (0, 1)$, the smaller $\Delta$ indicates that the less effect of our methods on graph density. In the Table 9 we calculate $\Delta$ on AM-Photo, Snap-Patent and Arxiv-Year datasets. The results reveal that on the AM-Photo dataset, graph rewiring increases density by 10.3%, while on the Snap-Patent and Arxiv-Year datasets, the increases are only 6.7% and 3.2% respectively. These findings demonstrate that our rewiring method generally has a modest effect on graph density.

## E RELATED WORK

### E.1 DIGRAPH LAPLACIAN

While the Laplacian for undirected graphs has been extensively studied, the area of Laplace operator digraphs remains underexplored. Chung (2005) pioneers this area by defining a normalized Laplace operator specifically for strongly connected directed graphs with nonnegative weights. This operator is expressed as $\mathbf{I} - \frac{\pi^{1/2}\mathbf{P}\pi^{-1/2}+\pi^{-1/2}\mathbf{P}^*\pi^{1/2}}{2}$. Key to this formulation is the use of the transition probability operator $\mathbf{P}$ and the Perron vector $\pi$, with the operator being self-adjoint. Building on the undirected graph Laplacian, Singh et al. (2016) adapt this concept to accommodate the directed structure, focusing particularly on the in-degree matrix. They define the directed graph Laplacian as $\mathbf{D}_{\text{in}} - \mathbf{A}$, where $\mathbf{D}_{\text{in}} = \text{diag}\left(\{d_i^{\text{in}}\}_{i=1}^N\right)$ represents the in-degree matrix. Li & Zhang (2012) uses stationary probabilities of the Markov chain governing random walks on digraphs to define the Laplacian as $\pi^{\frac{1}{2}}(\mathbf{I}-\mathbf{P})\pi^{-\frac{1}{2}}$, which underscores the importance of random walks and their stationary distributions in understanding digraph dynamics. Hermitian Laplacian Furutani et al. (2020) consider the edge directionality and node connectivity separately, and encode the edge direction into the argument in the complex plane. Diverging from existing Laplacians, our proposed DiLap $\mathbf{B}\text{diag}\left(\{\mathbf{P}_{ij}\}_{(v_i,v_j)\in E}^M\right)\mathbf{B}^\top$ is grounded in graph signal processing principles, conceptualized as the divergence of a signal's gradient on the digraph. It encompasses the degree matrix $\mathbf{D}$ to preserve local connectivity, the transition matrix $\mathbf{P}$ to maintain the graph's directed structure, and the diagonalized Perron vector $\mathbf{\Pi}$, capturing critical global graph attributes such as node structural importance, global connectivity, and expected reachability (Chung, 1997).

### E.2 DIGRAPH NEURAL NETWORKS

To effectively capture the directed structure with GNNs, spectral-based methods (Zhang et al., 2021; Tong et al., 2020a;b) have been proposed to preserve the underlying spectral properties of the digraph by performing spectral analysis based on the digraph Laplacian proposed by (Chung, 2005). Koke &

Cremers (2024) introduce holomorphic filters as spectral filters for digraphs, and investigate their optimal filter-bank. MagNet (Zhang et al., 2021) and its extensions (Lin & Gao, 2023; Fiorini et al., 2023) utilizes magnetic Laplacian to derive a complex-valued Hermitian matrix to encode the asymmetric nature of digraphs. Spatial GNNs also offer a natural approach to capturing directed structures. For instance, GraphSAGE (Hamilton et al., 2017) allows for controlling the direction of information flow by considering in-neighbors or out-neighbors separately. DirGNN (Rossi et al., 2023) further extends this framework by segregating neighbor aggregation according to edge directions, offering a more refined method to handle the directed nature of graphs. Transformer-based methods capture directed structure by specific positional encoding modules, such as directional random walk encoding (Geisler et al., 2023) and partial order encoding (Luo et al., 2024). DUPLEX (Ke et al., 2024) utilizes Hermitian adjacency matrix decomposition for neighbor aggregation and incorporates a dual GAT encoder for modeling directional neighbors.

