# OpenReview forum: "Commute Graph Neural Networks"
_ICLR.cc/2025/Conference — ICLR 2025 Conference Withdrawn Submission_

### Official Review · Reviewer_EWXV · 2024-10-17

**Soundness:** 2
**Presentation:** 2
**Contribution:** 2
**Rating:** 5
**Confidence:** 4

**Summary:**

This paper proposes  Commute Graph Neural Networks (CGNN) for directed graphs, which is based on a new digraph Laplacian matrix taking into the commute time on a (possibly rewired) strongly connected graph. Theoretical and empirical analysis is provided.

**Strengths:**

1) The idea of considering commute time is novel and reasonable.
2) The topic of directed graph neural networks is important.
3) The source code is provided.

**Weaknesses:**

1) It is unclear why the proposed Laplacian is sparse. From Eq. (5), the matrix $P$ seems to be a complete matrix.
2) It is unclear what the relationship is between Eq. (5) and $D^{-2}L$ and why you should choose Eq. (5).
3) Being strongly connected is too strong an assumption, and it is not clear why the rewiring procedure only minimally alters the overall semantics of the original graph.
4) [1] mentions flow imbalance in directed graphs and is not discussed. It is also unclear whether the idea in [1] is considered unidirectional by the authors.
5) (minor) Grammar issues: e.g., line 115.5 "notations. We" should be "notations, we"

Reference:
 [1] He, Y., Reinert, G., & Cucuringu, M. (2022, December). DIGRAC: digraph clustering based on flow imbalance. In Learning on Graphs Conference (pp. 21-1). PMLR.

**Questions:**

1) Why is the proposed Laplacian sparse? From Eq. (5), the matrix $P$ seems to be a complete matrix.
2) What is the relationship between Eq. (5) and $D^{-2}L$ and why do you should choose Eq. (5)?
3) Why does the rewiring procedure only minimally alter the overall semantics of the original graph?

---

> ### Author Response · Authors · 2024-11-22
> **Response to Reviewer EWXV (part 1)**
>
> Thank you for the thoughtful feedback on our manuscript. We provide the following detailed responses to your major concerns.
>
> **Q1:** The footnote on page 1 is not sensible, and it is not clear why the previous methods are undirectional during shortest path computation. What do you mean by the footnote on page 1?
>
> **A1:** Thank you for your feedback. It appears there may have been a significant misunderstanding, particularly concerning the terminology used in our paper. To clarify the footnote on page 1: The term '**uni**directional' is used to describe relationships in directed graphs, where edges have a specific direction from one node to another. This is distinct from '**un**directional,' which implies that the edges do not have a specified direction. I acknowledge that these terms are quite similar and understand how this could lead to confusion. The correct understanding of these terms is crucial, as it directly impacts the interpretation of our research's motivation and framework.
>
> Why do current digraph neural networks only capture **uni**directional relationships between neighboring nodes?  To answer this question, let's consider the directed graph in Figure 1. The central node, $v_i$, has three one-hop neighbors: $v_m$ as an in-neighbor and  $v_j$  and $v_k$  as out-neighbors. $v_m$ is connected to $v_i$ via an incoming edge, while both  $v_j$ and $v_k$ are connected to $v_i$ through outgoing edges. State-of-the-art digraph neural networks, such as MagNet and DirGNN, utilize one-layer directed message passing to differentiate between incoming and outgoing neighbors of $v_i$.  Consequently, in the process of learning the representation of $v_i$, since both $v_j$ and $v_k$  share the same edge direction relative to $v_i$, these models treat them as structurally equivalent with respect to the central node.  In other words, within these frameworks, $v_j$ and $v_k$  are considered **uni**directionally equivalent to $v_i$ because they are both one-hop out-neighbors.
>
> Why can our proposed CGNN model capture mutual relationships rather than just **uni**directional relationships between neighboring nodes? Although both $v_j$ and $v_k$ are one-hop out-neighbors of $v_i$ and therefore **uni**directionally equivalent, they differ significantly in their commute distances to $v_i$. Specifically, $v_j$ requires 5 hops to return to $v_i$, whereas  $v_k$ only needs 3 hops to return to $v_i$ based on the directed nature of the graph. Despite their **uni**directional equivalence, these differing commute distances suggest varying strengths in their relationships with $v_i$, i.e.,  nodes with shorter commute distances are deemed to have stronger relationships. Consequently, the relationship between $v_i$ and $v_k$ is stronger than that between  $v_i$ and $v_j$. Our research integrates these commute relationships between nodes into the graph neural network model to better reflect the complexity of real-world interactions.
>
> **Q2:** Why is the proposed Laplacian sparse? From Eq. (5), the matrix $\mathbf{P}$ seems to be a complete matrix.
>
> **A2:** If the adjacency matrix $\mathbf{A}$ of a given graph is not complete, meaning the graph itself is not a complete graph, then its corresponding transition matrix  $\mathbf{P}$  will not be a complete matrix either. This follows because $\mathbf{P}= \mathbf{D}^{-1}\mathbf{A}$ , where the degree matrix $\mathbf{D} \in \mathbb{R}^{N \times N}$ is a diagonal matrix whose diagonal elements represent the degrees of the nodes. Therefore, any zero entry in  $\mathbf{A}$ will result in a corresponding zero entry in  $\mathbf{P}$. In most real-world graphs,  $\mathbf{A}$ is typically sparse, which consequently makes  $\mathbf{P}$ a sparse matrix as well.
>
> **Q3:** What is the relationship between Eq. (5) and $D^{-2}L$ and why do you should choose Eq. (5)?
>
> **A3:** We apologize for the confusion in the definition and derivation of Eq.(5), i.e., the directed graph Laplacian (DiLap), in our initial submission. We have update the the definition of DiLap in Section 4.1 and its derivation in Appendix A.1. The DiLap defined in Eq. (5) does not have any relationship with $D^{-2}L$. For undirected graphs, the graph Laplacian is defined as the divergence of the gradient of a graph signal [1,2], which acts as a smoothness operator on the graph. Inspired by this, we define the divergence of the gradient for signals on directed graphs, as detailed in Appendix A.1. This definition allows the DiLap to serve similarly as a smoothness operator on directed graphs, demonstrating its rationality and adherence to the essence of the traditional graph Laplacian concept.

---

> ### Author Response · Authors · 2024-11-22
> **Response to Reviewer EWXV (part 2)**
>
> **Q4:** Why does the rewiring procedure only minimally alters the overall semantics of the original graph?
>
> **A4:** Our proposed similarity-based rewiring method involves two steps: initially generating a line graph, as shown in  $G^\prime$ in Fig.2,  where $G^\prime$ is a strongly connected graph with each node having at most two edges. We then combine  $G^\prime$ and $G$. This approach minimally alters the main structural semantics of the original graph, because each node in the original graph at most be added at most two additional edges.  To quantify the alterations brought from graph rewiring, we define edge density as  $\delta = \frac{M}{M_{\text{max}}}$, where $M_{\text{max}}$ is the maximum possible number of edges ($N^2$ for both $G$ and $\widetilde{G}$) in the graph and $M$ is the actual number of edges. We denote the edge density of the original graph $G$ as $\delta$ and that of the rewired graph  $\widetilde{G}$ as $\widetilde{\delta}$. Thus the change of graph density after rewiring can be represented as $\Delta = \frac{\widetilde{\delta} - \delta}{\delta} \in (0, 1)$, the smaller $\Delta$ indicates that the less effect of our methods on graph density. In the below table we calculate $\Delta$ on AM-Photo, Snap-Patent and Arxiv-Year datasets. The results reveal that on the AM-Photo dataset, graph rewiring increases density by 10.3%, while on the Snap-Patent and Arxiv-Year datasets, the increases are only 6.7% and 3.2% respectively. These findings demonstrate that our rewiring method generally has a modest effect on graph density.
>
> |          | AM-Photo | Snap-Patent | Arxiv-Year |
> | -------- | -------- | ----------- | ---------- |
> | $\Delta$ | 0.103    | 0.067       | 0.032      |
>
> **Q5:** [3] mentions flow imbalance in directed graphs and is not discussed. It is also unclear whether the idea in [1] is considered undirectional by the authors.
>
> **A5:** Thank you for bringing the DIGRAC to our attention. Upon thorough review of the literature, we confirm that DIGRAC primarily captures **uni**directional relationships between nodes due to its reliance on the standard message passing framework. This framework inherently focuses on **uni**directional interactions among nodes. However, it is important to note that DIGRAC's main objective is to enhance node clustering through a cluster-aware self-supervised loss, rather than to address commute time between nodes.
>
> In contrast, our CGNN model is specifically designed to capture and utilize commute times, offering a unique perspective by integrating this aspect into the graph neural network. This capability allows CGNN to account for varying path lengths and directions in directed graphs, which is not addressed by the DIGRAC framework. This distinction is crucial for understanding the specific contributions and focus of our work compared to that cited in the literature.
>
> **Q6:** Grammar issues: e.g., line 115.5 "notations. We" should be "notations, we"
>
> **A6:** Thank you for pointing out the grammar issue. We will correct it in the next version of the manuscript.
>
> [1] The emerging field of signal processing on graphs: Extending high-dimensional data analysis to networks and other irregular domains. IEEE signal processing magazine, 2013
>
> [2] Graph representation learning. Morgan & Claypool Publishers, 2020
>
> [3] DIGRAC: digraph clustering based on flow imbalance. In Learning on Graphs Conference (pp. 21-1). PMLR.

---

> ### Comment · Reviewer_EWXV · 2024-11-22
>
> I apologize for missing the letter "i" in your footnote. I have removed all comments in that regard and adjusted my score accordingly.
>
> Regarding the statement of being sparse for P, I think your current statement in the paper needs to be modified to reflect that P is not a complete matrix.

---

> > ### Author Response · Authors · 2024-11-23
> > **Response to the follow-up question**
> >
> > Thank you for your feedback. To clarify the sparsity of the matrix $\mathbf{P}$, we have revised the text in Section 2 of our paper. The updated statement now reads:  'Given that $\mathbf{D}^{-1}$ is a diagonal matrix and considering that real-world graphs are typically sparse ($M \ll N^2$), $\mathbf{A}$ and consequently $\mathbf{P}$ can generally be considered sparse.' This modification has been highlighted in blue in the new version of our paper.

---

> > > ### Comment · Reviewer_EWXV · 2024-11-28
> > >
> > > I hereby acknowledge reading your further response.

---

> ### Author Response · Authors · 2024-11-25
> **Looking forward to your reply**
>
> Dear Reviewer EWXV
>
> Thank you once again for your thorough and insightful feedback. We have endeavored to address all your concerns in our responses. As we near the end of this discussion phase, we are eager to know if our explanations have satisfactorily addressed your points.
>
> If you have any more comments or questions about our rebuttal, we strongly welcome your feedback. Your guidance is crucial for improving our work, and we look forward to any additional thoughts you might have.
>
> Warm regards,
> The Authors of Submission 6734.

---

> ### Author Response · Authors · 2024-11-27
> **Looking forward to your further reply**
>
> Dear Reviewer EWXV,
>
> As the discussion deadline approaches, we have uploaded a revised version of our manuscript that fully incorporates all your suggestions and has been thoroughly polished. Specifically, we have made the following improvements based on your feedback:
>
> * In Section 2, we provide a detailed explanation demonstrating the sparsity of the transition probability matrix $\mathbf{P}$.
> * In Appendix D.5, we present additional experiments to illustrate how the rewiring procedure influences the overall semantics of the original graph.
> * We have corrected all grammatical issues and typos.
>
> We look forward to your response and are eager to address any further questions you may have.
>
> Best regards,
>
> All authors

---

### Official Review · Reviewer_m8Kb · 2024-10-31

**Soundness:** 3
**Presentation:** 4
**Contribution:** 3
**Rating:** 8
**Confidence:** 4

**Summary:**

The authors present a novel approach that integrates node-wise commute time into a message-passing framework, introducing the Commute Graph Neural Network (CGNN). The central contribution of CGNN is the development of a new directed graph Laplacian, designed to address path asymmetry in directed graphs. The authors demonstrate that CGNN outperforms existing baselines across most datasets and effectively motivates the significance of the problem they address.

Overall, I found the paper well-executed and recommend it for acceptance.

**Strengths:**

This paper is engaging and well-written, with a thorough background review that enhances accessibility and readability. Key strengths I noted include:

1. **Novel Approach with Significant Potential:** The proposed method, particularly the newly formulated digraph Laplacian, offers a fresh perspective with substantial potential for future research and applications.

2. **Comprehensive Component Analysis (Section 5.3):** The inclusion of a component analysis strengthens the paper by providing an effective ablation study.

3. **Clear Contribution and Baseline Comparison:** The authors clearly articulate their contributions, outlining the distinctions between their method and existing baselines. They explain where prior approaches fall short and demonstrate how their approach addresses these limitations.

4. **Effective Visual Aids:** Figures 1 and 2 are well-designed and enhance understanding by clarifying details within the method.

5. **Robust Experimental Validation:** he paper validates its approach across a wide variety of datasets and multiple baseline comparisons, highlighting the robustness and generalizability of the proposed method.

6. **Reproducibility:** The authors provide code for reproducing their experiments.

**Weaknesses:**

Overall, this paper is strong in its methodology and results, though I have a few recommendations that could enhance its clarity and depth.

1. **Graph Density in Rewiring Approach:** While I appreciate that the authors provided commute times before and after rewiring in Table 3, it would be beneficial to also examine how rewiring affects graph density. This additional metric could offer deeper insights into structural changes post-rewiring.

2. **Unobserved Edges in Definition of $m_{i,in}^{(l)}$ and $m_{i,out}^{(l)}$:** Given that unobserved edges are introduced to the graph, I suggest adjusting the definitions of $m_{i,in}^{(l)}$ and $m_{i,out}^{(l)}$  to account for these edges, potentially assigning them a lower weight than observed edges. This adjustment could yield a more realistic representation of edge significance.

3. **Model Complexity:** The model’s complexity is relatively high, even though it’s reported to be on par with other GNN models. This complexity, particularly in precomputation, might be a barrier in some cases. However, I do not consider this a critical issue, as future work could address and optimize this aspect.

4. Inclusion of Synthetic Datasets: While the paper impressively covers a range of empirical datasets, the addition of synthetic datasets could improve interpretability. By embedding known patterns, synthetic data could highlight the model's strengths and limitations in detecting specific features.

5. Reordering Related Work: Placing the Related Work section (currently Section 6) closer to the beginning would make the reading experience smoother, giving readers essential context before diving into the methodology and results.

These revisions would, in my opinion, strengthen the paper without diminishing its core contributions.

**Questions:**

I have no further questions, though I would recommend that the authors address the previously noted weaknesses.

---

> ### Author Response · Authors · 2024-11-22
> **Response to Reviewer m8Kb**
>
> We sincerely appreciate the reviewer's constructive feedback and  positive remarks on our work. We provide the following detailed responses to your major concerns. We will add all revisions into the next version of our paper.
>
> **Q1:** It would be beneficial to also examine how rewiring affects graph density.
>
> **A1:**  We appreciate your suggestion to examine how our rewiring strategy affects graph density. Our approach, as illustrated in Figure 2, involves a two-step process: initially, we generate a line graph $G^\prime$, , which is a strongly connected graph where each node has at most two edges. We then merge $G^\prime$ with the original graph $G$  to create the rewired graph $\widetilde{G}$. This method minimally alters the main structural semantics of the original graph, as it adds at most two additional edges per node. To quantify these changes, we define edge density as  $\delta = \frac{M}{M_{\text{max}}}$, where $M_{\text{max}}$ is the maximum possible number of edges ($N^2$ for both $G$ and $\widetilde{G}$) in the graph and $M$ is the actual number of edges. We denote the edge density of the original graph $G$ as $\delta$ and that of the rewired graph  $\widetilde{G}$ as $\widetilde{\delta}$. Thus the change of graph density after rewiring can be represented as $\Delta = \frac{\widetilde{\delta} - \delta}{\delta} \in (0, 1)$, the smaller $\Delta$ indicates that the less effect of our methods on graph density. In the below table we calculate $\Delta$ on AM-Photo, Snap-Patent and Arxiv-Year datasets. The results reveal that on the AM-Photo dataset, graph rewiring increases density by 10.3%, while on the Snap-Patent and Arxiv-Year datasets, the increases are only 6.7% and 3.2% respectively. These findings demonstrate that our rewiring method generally has a modest effect on graph density.
>
> |          | AM-Photo | Snap-Patent | Arxiv-Year |
> | -------- | -------- | ----------- | ---------- |
> | $\Delta$ | 0.103    | 0.067       | 0.032      |
>
> **Q2:** Unobserved Edges in Definition of $m_{i,in}^{(l)}$ and $m_{i,out}^{(l)}$.
>
> **A2:** Thank you for your insightful suggestion regarding the treatment of unobserved edges. It is important to clarify that our model operates under the assumption that all edges within the graph data are observed, meaning that we have complete knowledge of all edges and their respective directions. The primary objective of our model is to leverage this complete edge information to learn node relationships effectively, utilizing directed edges with commute times to enhance our understanding of graph dynamics.
>
> **Q3:** Inclusion of Synthetic Datasets.
>
> **A3:** Thank you for this valuable suggestion, which enhances the interpretability of incorporating commute time into GNNs. Reviewer WWVp also emphasized this point. Thus, we generate synthetic graph data as follows: The dataset will be used for binary classification on synthetic directed graphs, consisting of 3,000 nodes divided evenly between two classes (1,500 nodes per class). The features of these classes are drawn from Gaussian distributions: $\mathcal{N}(0,1)$ for the first class and $\mathcal{N}(3,1)$ for the second. To construct the edges, nodes within the same class are connected with an edge with a probability of 0.2, while nodes from different classes have a much lower connection probability of 0.02. These connections are assigned random directions. Additionally, we specify a predefined commute path length range of [2, 7]. This method allows us to create a graph where each node has an asymmetric commute path with its neighbors, facilitating a detailed examination of how graph neural networks perform under varying structural conditions. On this graph, we first apply CGNN to learn node representations. Subsequently, we calculate the Mutual Information (MI) between the central node and its neighbors with short commute times, denoted as $\alpha_s$, and between the central node and its neighbors with long commute times, denoted as $\alpha_l$. If the average $\overline{\alpha}_s > \overline{\alpha}_l$, it confirms that our model effectively preserves commute relationships. In our experiments on such graphs, CGNN achieved $\overline{\alpha}_s = 13.2974$ and $\overline{\alpha}_l = 6.5521$, which aligns with our model's purpose.
>
> **Q4:** Reordering Related Work.
>
> **A4:** Thank you for your suggestion. In the revised version of the paper, we will incorporate your feedback and reorganize the structure of the paper accordingly.

---

> > ### Comment · Reviewer_m8Kb · 2024-11-25
> > **Response to authors comments**
> >
> > Thank you for clarifying and thoroughly considering our reviews. I choose to keep my score.

---

### Official Review · Reviewer_WWVp · 2024-11-03

**Soundness:** 2
**Presentation:** 3
**Contribution:** 2
**Rating:** 5
**Confidence:** 4

**Summary:**

In this paper, the authors propose a method for weighing the features of the neighbors of a node during aggregation step of GNN based on the commute distance of the neighbor to the node. The commute distance between nodes A and B is the average number of steps that a random walk takes traversing from node A to node B and back to node A. This distance is particularly relevant for directed graphs because although all nearest neighbors are one hop away from a node, their commute distances might vary because of the constraints imposed by the directions of the edges. One neighbor might require a circuitous path along many other nodes before returning to the original node (have longer commute distance) whereas another neighbor could be closer. The authors' key idea is to weight the importance of the features of the neighbors of a node during the aggregation step of GNN based on the commute distance of the node to those neighbors. Besides this weighing, the aggregation and update scheme that they use is based on that of Rossi et al. where the features of the incoming and outgoing neighbors are aggregated separately and used alongside the node's own features during the update step. The authors also propose an efficient way of computing commute distance. To do so, they introduce a weighted Laplacian for directed graphs that accounts both for the directional connectivity of the nodes and their importance (computed as the stationary probability at each node of a random walk on the graph). The authors also introduce a way to rewire a graph to ensure that it is irreducible and aperiodic while keeping the graph sparse (unlike alternative methods such as PageRank). The commute distance can then be efficiently computed using the sparse weighted Laplacian. Finally, the authors show empirically that their proposed approach improves on existing methods when applied to many standard directed graph data sets such as Squirrel and Chameleon.

**Strengths:**

This is an interesting paper with a clever insight. It is sensible to say that not all nearest neighbors on a directed graph are created equal. Weighing the features of some neighbors more during the aggregation step of a GNN based on the shorter commute distance of those neighbors to the original node is an intriguing idea. Weighing based on the commute distance certainly sounds reasonable. The proposed method for rewiring a graph to make it irreducible and aperiodic while still retaining sparsity is also clever as is the weighted Laplacian that can be used to efficiently compute the commute distance leveraging its sparsity using methods such as randomized truncated singular value decomposition.

The state-of-the-art performance achieved using the author's method on some of the most commonly used directed graph data sets, such as Squirrel, Chameleon, etc. is impressive and provides reasonable empirical proof of the validity of the proposed approach. The authors also include solid empirical evidence on running times of their algorithm and convincing comparisons to PageRank for graph rewiring and calculation of commute distances.

**Weaknesses:**

To this reviewer, the biggest weakness of the paper was that although weighing neighbors by commute time is sensible, it is not necessarily principled. Is there any reason to a priori expect that neighbors of a node that have shorter commute times to that node somehow contain more relevant features for learning on graphs? This seems to depend on the nature of the learning problem and the data set. Now, the authors can argue that their empirical evidence is sufficient to motivate their approach. However, more should be done here to support the author's proposal. Some evidence is provided in that an adjacency matrix constructed by weighing the neighbors by their commute distance more closely resembles an adjacency matrix constrained to edges that connect nodes within the same class. The authors should expand on this. What does this look like for other data sets? How does aggregation of information using these weighted neighbors look across multiple hops and longer distances across the graph? The authors should have come up with synthetic data sets that can elucidate the mechanism behind the improvement that they are seeing.

If empirical evidence is the main motivation behind the proposed schemes, the authors could have dome more to build a stronger case. An argument is made in the paper that with the weighing of the neighbors less irrelevant information is aggregated as the GNN models go deeper. The authors should empirically demonstrate this by showing how the performance of their model changes with depth and contrast with existing models. In general, it would have been very interesting to see the impact of the weights proposed in this paper on multi-hop GNN models, such MixHop, Shortest Path Networks, or DRew. In addition, it would have been more convincing if the authors had applied their approach to real world problems of directed graphs in addition to standard benchmark data sets used in Table 1 such as temporal web traffic data, power grids, traffic flow, etc.

The method proposed in the paper to rewire the graph to ensure irreducible and aperiodic graphs is also ad hoc and not very principled. The method certainly produces a sparse graph unlike PageRank, however, other approaches can also be used to generate irreducible graphs that are sparse such as generating a kNN graph based on the node features. It is not clear that the proposed method is optimal in any way other than that it outperforms the amended probabilities used in PageRank. See Questions below for suggestions on how the authors can evaluate this empirically.

**Questions:**

As outline in the weaknesses above, can the authors provide a principled reason for why features aggregated from a node's neighbors should be weighed by the commute distance of the node to its neighbors? Can the authors provide any theoretical justification for using commute distance for such weighing?

The authors provide one figure where they show for Chameleon and Squirrel data sets that an adjacency matrix constructed from nearest neighbors weighted by their commute distance more closely resembles an adjacency matrix constrained to connecting nodes within the same class. Can the authors include other empirical measures of how the weighting of neighbors can improve performance of GNNs? Can the authors look at properties of longer hops (going beyond one-hop neighbors) and how information is aggregated across nodes within or outside the same class? Would it be possible to construct a synthetic data set that would shed light on the mechanism behind why they see an improvement?

How does the proposed model's performance change with depth? The authors claim that the weighted neighbors avoids the problem of aggregation of irrelevant information as depth is increased? It would be informative to see an empirical demonstration of this. The authors should plot the performance of their model as a function of model depth and compare with existing models.

In addition, can the authors apply their approach a real world problem on directed graphs that requires long range information transmission, such as power grids or traffic flow data? This would bolster the empirical support for their method.

As noted in the weaknesses above, can the authors try alternative methods for graph rewiring that also produce sparse graphs, such as constructing a kNN graph using the node features? The authors should include an empirical comparison or provide theoretical justification for their method. Is the proposed similarity-based rewiring mode optimal?

How much of the improvement is coming from the fact that the Laplacian proposed by the authors is weighted by the stationary probability of random walks on the graph (or what the authors call the importance of a node)? Can the authors do an ablation study to disentangle this from the commute distance?

This reviewer was confused by the comment that general GNNs can outperform models tailored for directed graphs with hyper-parameter tuning. In Table 1, DirGNN, a model tailored for directed graphs, outperforms GCN. Can the authors clarify what they mean by this comment?

---

> ### Author Response · Authors · 2024-11-21
> **Response to Reviewer WWVp (Part 1)**
>
> We greatly appreciate your valuable time and constructive comments. We hope our answers can fully address your concerns.
>
> **Q1:** To this reviewer, the biggest weakness of the paper was that although weighing neighbors by commute time is sensible, it is not necessarily principled. Is there any reason to a priori expect that neighbors of a node that have shorter commute times to that node somehow contain more relevant features for learning on graphs? This seems to depend on the nature of the learning problem and the data set. As outline in the weaknesses above, can the authors provide a principled reason for why features aggregated from a node's neighbors should be weighed by the commute distance of the node to its neighbors?  Can the authors provide any theoretical justification for using commute distance for such weighing?
>
> **A1:** We recognize that the rationale for incorporating commute time in GNNs is based more on empirical observations and intuitive reasoning than on established principles. We also acknowledge that incorporating commute times, which highlight mutual relationships between nodes, may **not always** help and sometimes provide only marginal benefits. For example, on the *SNAP-PATENTS* and *CoraML* dataset, we observed that adding commute time-based weights during  message passing did not significantly enhance performance. Now we can  analyze the reason from the perspective of dataset.  *CoraML* is a directed citation network where nodes predominantly link to other nodes within the same research area. However, in such networks, reciprocal citations between two papers are impossible due to their chronological sequence. Consequently, **mutual path dependencies do not exist**, and thus, incorporating commute times to adjust neighbor weights might (slightly) hurt performance. A similar situation exists with the *SNAP-PATENTS* dataset, where each directed edge represents a citation from one patent to another, again indicating the absence of mutual path dependencies.
>
> However, in many real-world directed graph scenarios, the use of commute times proves beneficial. Our experiments, as illustrated in Figures 3 and 6, demonstrate the value of commute time in enhancing GNN performance across various datasets. The results indicate that leveraging commute time to weigh neighbor interactions can effectively help the model filter out irrelevant heterophilic information, thereby improving the relevance and quality of the information propagated during message passing.
>
> Although the utility of commute time is conditional and our approach may not be always applicable, we believe that our work contributes new insights to the analysis of directed graphs. Our model is pioneering in its identification and handling of mutual path dependencies in directed graphs, which is vital for representing real-world relationships between entities—an aspect largely neglected in previous research. This represents a contribution to the directed graph analysis community and underscores the innovative nature of our study.

---

> ### Author Response · Authors · 2024-11-21
> **Response to Reviewer WWVp (Part 2)**
>
> **Q2.1:** Can the authors include other empirical measures of how the weighting of neighbors can improve performance of GNNs?
>
> **A2.1:** Thank you for your suggestions. For the first question, as indicated in Section 5.1, Figure 3, we assess the effectiveness of commute time in enhancing message passing by comparing the squared Frobenius norm of differences between the label similarity matrix, $\mathcal{M}$, and two propagation matrices: the commute-time-based propagation matrix $\widetilde{\mathcal{C}}^{\text{in}} + \widetilde{\mathcal{C}}^{\text{out}}$, and the original propagation matrix $\mathbf{A}+\mathbf{A}^\top$. This comparison helps to illustrate how commute time aids in filtering out irrelevant heterophilic information during message passing. o further address your query regarding the weighting of neighbors and its impact on GNN performance, we have expanded our analysis in Appendix D.3, Figures 6(a) through 6(e). In this section, we compare our proposed commute-time-based message passing with the propagation mechanisms used in GCN and PPRGo. This comparison is designed to showcase our model's proficiency in managing both homophilic and heterophilic graphs, and its ability to effectively filter noise from neighbors.
>
> To address the request for other empirical measures of neighbor re-weighting, we propose utilizing **Mutual Information (MI)** as a metric. This measure would compute the mutual information between the aggregated features of neighbors (after weighting) and the target labels. A higher mutual information between the representations of neighboring nodes with the same label suggests that the weighted aggregation effectively captures more relevant information for the prediction task. This approach could provide an understanding of how well the neighbor weighting strategy enhances the performance of the GNN in various learning scenarios. Specifically, given the node representation $\mathbf{Z}$ learned by the CGNN with re-weighted neighbors, and compare it to $\mathbf{Z}^\prime$ , the representation learned by a standard GCN without edge re-weighting.  We define the average MI between a central node $v_i$  and its homophilic neighbors  $\mathcal{N}^{\text{homo}}\_i$ as $\delta_i = \frac{1}{|\mathcal{N}^{\text{homo}}\_i|}\sum_{v_j \in \mathcal{N}^{\text{homo}}\_i}\mathrm{MI}(\mathbf{Z}\_i, \mathbf{Z}\_j)$ for CGNN with edge re-weighting. Considering all $N$ nodes, the average MI across the graph can be expressed as $\overline{\delta} = \frac{1}{N}\sum_i \delta_i$. Similarly, for GCN without edge re-weighting, the MI between homophilic neighboring nodes is represented as  $\delta^\prime_i = \frac{1}{|\mathcal{N}^{\text{homo}}\_i|}\sum_{v_j \in \mathcal{N}^{\text{homo}}\_i}\mathrm{MI}(\mathbf{Z}\_i^\prime, \mathbf{Z}\_j^\prime)$, and the graph average MI is denoted as $\overline{\delta}^\prime = \frac{1}{N}\sum_i \delta^\prime_i$. Our experiments on directed graph datasets in the below table demonstrate that our proposed commute-time-based edge re-weighting method more effectively captures relevant information from homophilic neighbors, thereby enhancing the model’s predictive accuracy for tasks involving nodes with similar labels.
>
> |                            | Squirrel   | Chameleon   | Citeseer    | AM-Photo    |
> | -------------------------- | ---------- | ----------- | ----------- | ----------- |
> | $\overline{\delta}$        | **9.5289** | **17.5234** | **25.0918** | **19.9649** |
> | $\overline{\delta}^\prime$ | 3.3130     | 9.5011      | 19.2517     | 17.0240     |

---

> ### Author Response · Authors · 2024-11-21
> **Response to Reviewer WWVp (Part 3)**
>
> **Q2.2:** Can the authors look at properties of longer hops and how information is aggregated across nodes within or outside the same class?
>
> **A2.2:** For the second question, your suggestion to explore the effects of longer hops and the aggregation of information across nodes within or outside the same class is compelling. In our proposed CGNN model, similar to many existing GNNs, the number of layers—which corresponds to the number of hops to be aggregated—is treated as a hyperparameter. This hyperparameter is adjusted based on the characteristics of different datasets to optimize performance. Our empirical findings, as detailed in Table 6, show that for homophilic datasets such as Citeseer and AM-Photo, fewer hops are generally sufficient to yield the best results. In contrast, for heterophilic datasets like Squirrel and Chameleon, where neighboring nodes often belong to different classes, our model benefits from more layers to effectively retrieve useful information from longer hops. To quantitatively assess the impact of varying hops on learning node representations, we introduced the metric $\overline{\delta}\_{\text{heter}}$ , which measures the average Mutual Information (MI) between the central nodes' representations and their heterophilic neighbors (those outside their class). We then calculate the ratio $\frac{\overline{\delta}}{\overline{\delta}\_{\text{heter}} + \overline{\delta}}$ to quantify the relative contribution of homophilic (within the same class) versus heterophilic (outside the same class) information in the node representation. The following table indicate that in heterophilic graphs, incorporating more hops allows the model to capture more useful information from broader neighborhood contexts. Conversely, fewer hops are generally sufficient in homophilic graphs to achieve optimal learning outcomes.
>
> | $\frac{\overline{\delta}}{\overline{\delta}_{\text{heter}} + \overline{\delta}}$ | Squirrel | Chameleon | Citeseer | AM-Photo |
> | -| - | - | - | - |
> | hop 1| 0.2103| 0.2517| 0.7736   | 0.8923   |
> | hop 3 | 0.6912| 0.5312| 0.7528   | 0.8917   |
> | hop 5| 0.7421| 0.6057| 0.6601 | 0.7764 |
>
> **Q2.3:** Can the authors look at properties of longer hops (going beyond one-hop neighbors) and how information is aggregated across nodes within or outside the same class?
>
> **A2.3:** For the **synthetic dataset**, we propose the following generation process: The dataset will be used for binary classification on synthetic directed graphs, consisting of 3,000 nodes divided evenly between two classes (1,500 nodes per class). The features of these classes are drawn from Gaussian distributions: $\mathcal{N}(0,1)$ for the first class and $\mathcal{N}(3,1)$ for the second. To construct the edges, nodes within the same class are connected with an edge with a probability of 0.2, while nodes from different classes have a much lower connection probability of 0.02. These connections are assigned random directions. Additionally, we specify a predefined commute path length range of [2, 7]. This method allows us to create a graph where each node has an asymmetric commute path with its neighbors, facilitating a detailed examination of how graph neural networks perform under varying structural conditions. On this graph, we first apply CGNN to learn node representations. Subsequently, we calculate the Mutual Information (MI) between the central node and its neighbors with short commute times, denoted as $\alpha_s$, and between the central node and its neighbors with long commute times, denoted as $\alpha_l$. If the average $\overline{\alpha}\_s > \overline{\alpha}\_l$, it confirms that our model effectively preserves commute relationships. In our experiments on such graphs, CGNN achieved $\overline{\alpha}\_s = 13.2974$ and $\overline{\alpha}\_l = 6.5521$, which aligns with our model's purpose.
>
> **Q3:** How does the proposed model's performance change with depth?
>
> **A3:** Thank you for your insightful observation. Our model is designed to accurately capture the strength of relationships between **neighboring nodes**. However, we agree that investigating model's capacity to handle oversmoothing is also interesting. Thus, we have conducted experiments to assess this. Specifically, we tested the CGNN with varying depths of 1, 3, 5, and 10 layers on the AM-Photo dataset and compared the results with those of a standard GCN and DirGNN. These experiments demonstrate how our model suffer from oversmoothing as it increases in depth. The results indicate that while GCN, DirGNN, and CGNN all exhibit some degree of oversmoothing, CGNN consistently outperforms the baseline models even as the number of layers increases.
>
> | # Layers | 1 | 3  | 5 | 10 |
> | - | - | - | - | - |
> | GCN   | 87.17| 87.03 | 83.53  | 76.33  |
> | DirGNN | 88.26| 87.92 | 84.96   | 75.92  |
> | CGNN | **90.01** | **90.29** | **88.24** | **78.85** |
>
> We will add the whole experiment including more baselines and datasets in the revised version of our paper.

---

> ### Author Response · Authors · 2024-11-21
> **Response to Reviewer WWVp (Part 4)**
>
> **Q4:** Can the authors apply their approach a real world problem on directed graphs that requires long range information transmission, such as power grids or traffic flow data?
>
> **A4:** We believe that our model has the potential to address real-world problems on directed graphs. However, specific domains such as traffic flow involve temporal graph data, which would require further adaptations or the addition of tailored modules to effectively apply CGNN to these contexts. In our current research, we have utilized datasets such as middle-scale datasets like AM-Photo and Squirrel, and large-scale datasets like Snap-Patents and Roman-Empire, which are commonly used in the literature on directed graph neural networks. These applications demonstrate the versatility of our model across various scales and types of directed graphs, providing a foundation for future extensions to more complex scenarios like power grids or dynamic traffic networks.
>
> **Q5:** Can the authors try alternative methods for graph rewiring that also produce sparse graphs, such as constructing a kNN graph using the node features? The authors should include an empirical comparison or provide theoretical justification for their method. Is the proposed similarity-based rewiring mode optimal?
>
> **A5:** We opted not to use kNN for sparsifying the graphs for two main reasons: First and the most intrinsic problem of kNN graph is that kNN graph inherently do not guarantee irreducibility. In the other words, kNN graph constructed based on node features may not be strongly connected, thus we can not compute meaningful and deterministic commute time based on kNN graph. Secondly, reconstructing the graph using kNN fundamentally alters the original structure of the graph. This significant change can adversely affect the performance on downstream prediction tasks, as it may discard important structural information inherent to the original graph.  In contrast, our proposed similarity-based rewiring method involves two steps: initially generating a line graph, as shown in  $G^\prime$ in Fig.2,  where $G^\prime$ is a strongly connected graph with each node having at most two edges. We then combine  $G^\prime$ and $G$. This approach minimally alters the main structural semantics of the original graph, because each node in the original graph at most be added two more edges. Moreover, because $G^\prime$ is irreducible, thus the rewired graph $\widetilde{G}$ also retains irreducibility. Thus we can use our proposed similarity-based rewiring strategy to compute meaningful and deterministic commute time. These reasons underscore why our similarity-based rewiring strategy is more suitable for our objectives than using a kNN-based method.
>
> **Q6:** How much of the improvement is coming from the fact that the Laplacian proposed by the authors is weighted by the stationary probability of random walks on the graph? Can the authors do an ablation study to disentangle this from the commute distance?
>
> **A6:** In our study, the integration of node importance (stationary probability) into the DiLap operator is aimed not only at enriching the structural information captured by DiLap but also at facilitating a simplified formulation of the fundamental matrix  $\mathbf{Z}$ (as detailed in the proof of Lemma 4.1 in Appendix A3).  However, we recognize the value of your suggestion to conduct an ablation study to disentangle this from the commute distance. To this end, we introduce $\mathrm{CGNN}\_{\text{un}}$, which utilizes an unweighted version of the DiLap operator. We will compare the node classification results between  $\mathrm{CGNN}\_{\text{un}}$ and the original CGNN model to specifically assess the impact of weighting by the stationary probability on the performance. We show the experimental results as follows:
>
> |        | Squirrel | Chameleon | Citeseer |
> | ---- | ---- | ----- | ------ |
> | $\mathrm{CGNN}\_{\text{un}}$ | 76.93    | 79.49     | 70.31    |
> | $\mathrm{CGNN}$             | 77.61    | 79.54     | 70.27    |
>
> **Q7:** This reviewer was confused by the comment that general GNNs can outperform models tailored for directed graphs with hyper-parameter tuning. In Table 1, DirGNN, a model tailored for directed graphs, outperforms GCN. Can the authors clarify what they mean by this comment?
>
> **A7:** We apologize for the confusion caused by our previous statement. Our intention was to convey that, with careful hyper-parameter tuning, general GNNs can achieve results comparable to, or even better than, **some of** GNNs tailored for digraphs (DiGCN, MagNet and DiGCL), as evidenced in the Squirrel, Chameleon, and AM-Photo datasets. While it is true that DirGNN outperforms GCN, our results in Table 1 of our paper shows that other directed graph models like DiGCN, MagNet, and DiGCL do not always perform better than a well-tuned vanilla GCN such as on Squirrel dataset. We have revised this statement in the latest version of our paper.

---

> ### Author Response · Authors · 2024-11-25
> **Looking forward to your reply**
>
> Dear Reviewer WWVp,
>
> We would like to express our deep gratitude for your comprehensive and insightful review of our paper. In response to the points you raised, we have provided detailed, point-by-point explanations and additional experiments to address your concerns thoroughly.
>
> Since the discussion due is approaching, would you mind checking the response to confirm where you have any further questions?
>
> We are looking forward to your reply and happy to answer your further questions.
>
> Warm regards,
>
> The Authors of Submission 6734.

---

> ### Author Response · Authors · 2024-11-28
> **A Kind Reminder to Reviewer WWVp**
>
> Dear Reviewer WWVp,
>
> Thank you once again for your insightful feedback on our submission. We would like to remind you that the discussion period is concluding. Considering the borderline score given during the initial review, your final decision is very crucial to the fate of our paper. Thus, We kindly urge you to review our responses.
>
> Below, we reiterate the key elements of our response to your comments, hoping to ensure that all your concerns have been thoroughly addressed.
>
> * We clarified the non-principled nature of our method and defined the specific application scope of our work.
> * We introduced a new empirical measure for re-weighting neighbor interactions.
> * We carried out additional experiments to explore the effects of longer hops and varying model depths.
> * We proposed and detailed a synthetic dataset to demonstrate how commute time facilitates the filtering of heterophilic information by our model.
> * We discussed why kNN-based rewiring cannot substitute for our proposed similarity-based rewiring technique.
> * We conducted further experiments involving different measures of node importance.
>
> We are eager to confirm whether our responses have adequately addressed your concerns. We look forward to any additional input you may provide.
>
> Warm regards,
>
> The Authors of Submission 6734.

---

> > ### Comment · Reviewer_WWVp · 2024-11-28
> >
> > I thank the authors for their thoughtful and comprehensive response. I also commend the authors for including new experiments and empirical support in their replies. However, I am still concerned about the lack of a principled justification for weighing neighboring nodes based on commute distance during aggregation of local information. I agree with the authors that their empirical results on benchmarking and standard datasets such as AM-Photo and Snap-Patents are very promising. However, I still think that this paper will benefit significantly from solving a real-world problem. Furthermore, the additional measurements provided by the authors, such as mutual information, the ratio of mutual information of heterophilic and homophilic neighbors, and the performance as a function of model depth, although interesting, provide no additional insight into why the proposed weights are useful. Therefore, I will stick with my original score.

---

### Official Review · Reviewer_rah7 · 2024-11-04

**Soundness:** 2
**Presentation:** 3
**Contribution:** 3
**Rating:** 5
**Confidence:** 4

**Summary:**

In the submitted manuscript, the authors propose a novel digraph Laplacian, which is later used to more efficiently calculate the commute time of pairs of nodes. They furthermore propose a simple node feature based rewiring scheme, which allows them to ensure that the resulting graph gives rise to an aperiodic, irreducible Markov Chain, which has a unique steady state. The authors then propose to calculate commute times on this rewired graph, to transform these commute times by taking the exponential function of this matrix and subsequently sparsifying it with the adjacency matrix. This then allows the authors to propose a variant of the DirGNN, called CGNN, in which edges are reweighted by their transformed commute times. The authors finally evaluate the empirical performance of their CGNNs against a large variety of baseline models on a large number of datasets and find consistently good, although sometimes marginal performance improvements. They furthermore analyse these results and provide several insightful further experiments on runtimes and ablation studies of different model components.

**Strengths:**

- The rewiring scheme that you propose is simple, but rather nice in my opinion. It would be interesting to see further study of its impact on the overall graph structure.

- Your proposed CGNNs are compared to a comprehensive set of baseline models, which great to see.

- The analysis of the application scope of your proposed model is very strong indeed and something that is generally not done enough in our literature.

**Weaknesses:**

Please find further details on my listed weaknesses in my questions below.
- The proposed model boils down a weighting of the DirGNN by an efficiently calculated function of the commute times, which is a rather trivial change.
- The derivation of the DiLap matrix appears to be flawed.
-  Your ablation studies could be improved and extended.

**Questions:**

1] The derivation of your DiLap operator appears to be flawed. In particular, in the second line of Equation (12) when you pull $s_i$ out of the sum, the term $s_i$ arises $d_i^{out}$ times and therefore, Line 2 should be $s_i - \frac{1}{d_i^{out}} \sum s_j.$ The correction of this error means that you should be working with the random walk Laplacian (see e.g. [1]), which would be far more intuitive. To me it seems that for it to be possible to accept this paper at ICLR, the derivation of your operator needs to be corrected and the subsequent experiments should be adjusted.

2] I am unsure what adjacency matrix you use in Line 314 to sparsify the matrix $\tilde{\mathcal{C}}.$ Are you using the adjacency matrix corresponding to the graph in which you have added the node feature similarity edges? And if not, could you hypothesise how severe the impact may be of calculating the commute times on a rewired graph and to then message pass with the original graph.

3] Your method appears to be relatively memory intensive. In particular, you seem to require the evaluation of the exponential function fo the dense matrix $\tilde{\mathcal{C}}$. Empirical evaluation of, not only the time, but also memory complexity of your method in comparison to your baseline methods would be very valuable.

4] The ablation study in Table 2 is very interesting! I think it should be extended in scope to also extend to homophilic datasets and to also include other commonly used message passing operators, such as the symmetrically normalised adjacency matrix used in the GCN and the PageRank matrix used in the PPrGo model [2].

5] It does not seem sensible to me to compare your sparisfied commute time based CGNN to the CGNN$_{ppr}$ using the dense PageRank matrix. It seems to be fairer to me to either compare dense versions of both matrices or sparse versions of both matrices. In particular, since you sparify your commute time matrix with the adjacency matrix, it would be interesting to compare your model to a PageRank-based scheme, where the PageRank matrix is also sparsified with the adjacency matrix.

6] Minor comments:

6.1] The abbreviation "SPD" is used in Line 45 before its definition.

6.2] The contributions of your paper are not explicitly listed.


[1] Von Luxburg, U., 2007. A tutorial on spectral clustering. Statistics and computing, 17, pp.395-416.

[2] Bojchevski, A., Gasteiger, J., Perozzi, B., Kapoor, A., Blais, M., Rózemberczki, B., Lukasik, M. and Günnemann, S., 2020, August. Scaling graph neural networks with approximate pagerank. In Proceedings of the 26th ACM SIGKDD International Conference on Knowledge Discovery & Data Mining (pp. 2464-2473).

---

> ### Author Response · Authors · 2024-11-21
> **Response to Reviewer rah7 (Part 1)**
>
> We are grateful for your insightful comments and positive feedback on our paper. Below are our detailed response. All modifications have been highlighted in blue in our revised manuscript.
>
> **Q1:** The derivation of your DiLap operator appears to be flawed.
>
> **A1:** We apologize for the errors in the definition and derivation of the directed graph Laplacian (DiLap) in our initial submission. We recognize that the random walk Laplacian does not adequately represent the divergence of the gradient on directed graphs. In the revised version of our paper, we have rigorously redefined the divergence of the gradient on digraphs as $\mathbf{T} = \mathcal{D}\mathcal{G}$, where $\mathcal{D}$ is the divergence operator and $\mathcal{G}$ is the gradient operator.  Specifically, $\mathcal{G}$ maps a signal defined on the nodes of the graph to a signal on the edges, with $ (\mathcal{G}s)\_{(v_i,v_j)} = \mathbf{P}\_{ij} (s_i - s_j)$ and $\mathcal{D}$ maps a signal defined on the edges back to a signal on the nodes, where $\left(\mathcal{D}(\mathcal{G}s)\right)\_i = \sum\_{v_j \in \mathcal{N}\_i^{\text{in}}} (\mathcal{G}s)\_{(v_j,v_i)} - \sum\_{v_j \in \mathcal{N}\_i^{\text{out}}} (\mathcal{G}s)\_{(v_i,v_j)}$. Consequently, the new DiLap can be represented in matrix form of the composed operator $\mathcal{D}\mathcal{G}$ as $\mathbf{T} = \mathbf{B} \mathrm{diag}\left(\left\\{\mathbf{P}\_{ij}\right\\}\_{(v_i, v_j) \in E} ^ M \right) \mathbf{B}^\top$, with $\mathbf{B}$ serving as the incidence matrix.
>
> We have detailed these corrections and elaborated on the derivation in Appendix A of the revised manuscript. The respective changes have been highlighted in blue in Section 4.1. Additionally, we have updated the experimental results to reflect these adjustments. We appreciate your feedback and believe these modifications have significantly strengthened the paper.
>
> **Q2:** Are you using the adjacency matrix $\widetilde{\mathcal{C}}$ corresponding to the graph in which you have added the node feature similarity edges? Could you hypothesise how severe the impact may be of calculating the commute times on a rewired graph and to then message pass with the  original graph.
>
> **A2:** Thank you for pointing out the ambiguity in the sparsification of $\widetilde{\mathcal{C}}$.  In our work, the adjacency matrix used for sparsifying $\widetilde{\mathcal{C}}$ originates from the original graph $\mathbf{A}$, not the rewired one. The purpose of the similarity-based graph rewiring is solely to ensuring that the new graph remains both irreducible and aperiodic. It allows us to compute meaningful (non-zero) and deterministic node-wise commute times with the rewired graph. We then leverage these computed commute times to strengthen node relationships between **neighboring nodes in the original graph**, enhancing and refining the information flow during message passing. Thus, we use original adjacency matrix  $\mathbf{A}$ to sparsify $\widetilde{\mathcal{C}}$, which can filter the relations in the original graph.
>
> For the second question, we address this both intuitively and empirically. Intuitively, as detailed in Section 4.2 and illustrated in Figure 2, our similarity-based rewiring method introduces at most two additional edges per node, targeting those nodes with the highest feature similarity. This strategy is designed to minimally alter the original graph structure. Consequently, the overall results are likely similar whether using the original or the rewired graph. We here conduct empirical analyses which also support this assertion, indicating minimal impact on the outcomes due to the restrained scope of modifications. Specifically, we use the original adjacency matrix $\mathbf{A}$ and the rewired one $\widetilde{\mathbf{A}}$ to sparsify the $\widetilde{\mathcal{C}}$ respectively and report the average accuracy in the following table.
>
> |                          | Squirrel | Chameleon | AM-Photo |
> | ------------------------ | -------- | --------- | -------- |
> | $\mathbf{A}$             | 77.61    | 79.54     | 90.41    |
> | $\widetilde{\mathbf{A}}$ | 77.64    | 79.49     | 90.38    |

---

> ### Author Response · Authors · 2024-11-21
> **Response to Reviewer rah7 (Part 2)**
>
> **Q3:** Your method appears to be relatively memory intensive.
>
> **A3:** Thank you for your valuable feedback regarding the memory intensity of our method. Indeed, we totally acknowledge that the commute time matrix $\mathcal{C}$ is a dense matri, designed to preserve commute times between all pairs of nodes, resulting in a memory complexity that is quadratic with respect to the number of nodes, specifically $\mathcal{O} (N^2)$. In contrast, baseline methods such as GCN, GAT, and DirGNN primarily depend on memory proportional to the number of edges, $\mathcal{O}(|E|)$. This inherent difference underscores the increased memory demands of our method. On the other hand,  exponential function on $\mathcal{C}$ can be efficiently computed by first using $\mathbf{A}$ to sparsify it.
>
> Our complexity analysis in Section 4.4 demonstrates that the time complexity of our method is within a feasible range, consistent with most existing GNN models. We understand that memory cost is a crucial factor for scalability, especially on large graphs, and recognize that this remains a challenge for our CGNN. However, this challenge also presents a compelling opportunity for future research. We plan to explore this in our future work.
>
> **Q4:** The ablation study should be extended in scope to also extend to homophilic datasets and to also include other commonly used message passing operators, such as  the symmetrically normalised adjacency matrix used in the GCN and the  PageRank matrix used in the PPRGo model .
>
> **A4:** Thank you for your insightful suggestions. In response, we have included additional analyses in Figure 6 of Appendix D.3, which is highlighted in blue in the revised version of our paper. We conducted an ablation study on homophilic graphs, specifically analyzing label similarity in CoraML and Citeseer. The results, as shown in Figure 6a, confirm that our model also effectively filters and enhances useful information in these homophilic settings. Moreover, we have expanded our comparative framework by incorporating two other propagation matrices: $\widehat{\widetilde{\mathbf{A}}}$ from vanilla GCN, and the approximate personalized PageRank $\mathrm{APPR}$ from PPRGo. Figures 6b and 6d illustrate that while GCN and PPRGo manage to slightly reduce heterophilic information from neighbors during message passing, CGNN achieves significantly more substantial reductions. These findings underscore the robustness of CGNN in handling both homophilic and heterophilic information.
>
> **Q5:** It seems to be fairer to either compare dense versions of both matrices or sparse versions of both matrices.
>
> **A5:** This suggestion is valuable. Similar to our approach with the commute-time-based propagation matrix  $\mathcal{C}$ , where we use the adjacency matrix $\mathbf{A}$ to sparse it,, we have applied the same method to sparsify the PPR-based propagation matrix. The resulting model is denoted as  $\text{CGNN}\_{\text{sppr}}$. In the table below, we present a comparison of node classification results between  $\text{CGNN}$ and $\text{CGNN}\_{\text{sppr}}$.
>
> |                             | Squirrel | Chameleon | AM-Photo |
> | --------------------------- | -------- | --------- | -------- |
> | $\text{CGNN}$               | 77.61    | 79.54     | 90.41    |
> | $\text{CGNN}_{\text{sppr}}$ | 73.49    | 75.83     | 88.62    |
>
> **Q6.1:** The abbreviation "SPD" is used in Line 45 before its definition.
>
> **A6.1:** We have modified it to "shortest path distance (SPD)".
>
> **Q6.2:** The contributions of your paper are not explicitly listed.
>
> **A6.2:** Thank you for your feedback. Here we list the contributions of our work.
>
> (1)  We identify and address mutual path dependencies in directed graphs, which is crucial for representing real-world relationships between entities, a factor ignored in prior work. Further, we propose to use commute times to quantify the strength of node-wise mutual path dependencies.
>
> (2)  We extend the traditional graph Laplacian to directed graphs by introducing DiLap, a novel Laplacian based on signal processing principles tailored for digraphs. Leveraging DiLap, we develop an efficient and theoretically sound method for computing commute times that enhances computational feasibility.
>
> (3) We propose the Commute Graph Neural Networks (CGNN), which incorporate commute-time-weighted message passing into their architecture. Through comprehensive experiments across various digraph datasets, we demonstrate the effectiveness of CGNN.
>
> We have outlined the contributions of our work in the Introduction section, specifically from lines 88 to 99.

---

> > ### Comment · Reviewer_rah7 · 2024-11-25
> >
> > I want to thank the authors for their detailed response. The great majority of my concerns have been resolved (besides Points 1] and 3]).
> >
> > Concerning Point 3], it seems to me though that the large memory cost that you acknowledge in **A3** should also be mentioned in your revised manuscript, so as to be transparent about this limitation of your work.
> >
> > Regarding Point 1], I want to thank the authors very much for the comprehensive changes they have made. From what I could tell, the derivation of the new DiLap operator appears to be free of error. However, it is unclear to me, why you have chosen to work with this operator, which does not seem to have an obvious relation to the operator, that was discussed in your original submission. I also find your choice to not use the random walk Laplacian surprising, since it would have been the result of your previous derivation once corrected. The fact that it does not align with the concept of the "divergence of the gradient on digraphs", that you introduced in your revisions, seems to be insufficient justification to me. I think that it may be better for your work to go through another round of reviews with your new DiLap operator to make sure that it is well-motivated in the context of your work. I furthermore want to remark, that there is a minor error in your newly added Proposition 2: Your proof appears to show that $T$ is node permutation equivariant, not invariant as is currently claimed. I furthermore saw that while the results in Table 1 appear to have changed as a result of the operator change, the results in Tables 4 and 5 appear to remain the same.
> >
> > In general, I feel that your paper has improved during this rebuttal process. However, to me, your work does not seem ready for publication yet. Therefore, my original score "5: marginally below the acceptance threshold" still appears to be suitable to me and I choose to maintain it.

---

> > > ### Author Response · Authors · 2024-11-25
> > > **Response to follow-up questions**
> > >
> > > Dear Reviewer rah7,
> > >
> > > Thank you very much for your reply and follow-up questions. We address your questions as follows:
> > >
> > > **Q1:**  The large memory cost that you acknowledge in **A3** should also be mentioned in your revised manuscript.
> > >
> > > **A1:** We have included a detailed discussion of this limitation in **Section 6** of the revised manuscript to address the memory cost concerns.
> > >
> > > **Q2:** Why you have chosen to work with this operator, which does not seem to have an obvious relation to the operator, that was discussed in your original submission. I also find your choice to not use the random walk Laplacian surprising, since it would have been the result of your previous derivation once corrected. The fact that it does not align with the concept of the "divergence of the gradient on digraphs", that you introduced in your revisions, seems to be insufficient justification to me.
> > >
> > > **A2:** In the previous version of our manuscript, we defined the graph Laplacian simply as a smoothness operator for signals on directed graphs. While this approach led to a version of the random walk Laplacian, it did not adhere to the strict definition of "divergence of the gradient on graph signals," which forms the cornerstone of the undirected graph Laplacian concept [1]. In other words, this earlier version of DiLap essentially functioned as a general smoothness operator without fully encapsulating the divergence of the gradient on directed graph signals. In contrast, the revised DiLap presented in Eq. (5) and detailed in Appendix A.1, is founded on rigorously defined gradient and divergence operators specific to directed graphs. This formulation allows us to derive a DiLap that is not only more theoretically sound but also aligns more closely with the fundamental principles of graph signal processing.
> > >
> > > **Q3:** There is a minor error in your newly added Proposition 2: Your proof appears to show that $\mathbf{T}$ is node permutation equivariant, not invariant as is currently claimed.
> > >
> > > **A3:** Thank you for pointing out the error. We have corrected it in our updated version of the paper.
> > >
> > > **Q4:** The results in Tables 3 and 4 appear to remain the same.
> > >
> > > **A4:** We apologize for any confusion caused by the unchanged data in Tables 3 and 4. Due to the intensive nature of the rebuttal period, we initially focused on updating the results in the main experiments presented in Table 1. However, in the revised version of our paper, we have corrected and updated the results in Tables 3 and 4. Additionally, we have revised Figures 3, 4, and 5. You will find that Figures 3 and 4, although updated, display results that are closely similar to the previous findings.
> > >
> > > [1] The emerging field of signal processing on graphs: Extending high-dimensional data analysis to networks and other irregular domains. IEEE signal processing magazine, 2013.
> > >
> > > Thank you once again for your thoughtful review and the time you have invested in our paper. Please let us know if you have any further questions about our paper.
> > >
> > > Warm regards,
> > >
> > > All authors

---

> ### Author Response · Authors · 2024-11-25
> **Looking forward to your reply**
>
> Dear Reviewer rah7,
>
> Thank you once again for your insightful feedback on our submission. We would like to remind you that the discussion period is concluding. To facilitate your review, we have provided a concise summary below, outlining our responses to each of your concerns:
>
> * We have re-derived the directed graph Laplacian matrix based on the divergence of the gradient on the digraph signal, and adjusted the experiments.
>
> * We have conducted further experiments focused on graph rewiring, and performed an ablation study on message passing operators and the sparsified PPR.
>
> * We have made extensive revisions to enhance the clarity and polish of our paper.
>
> Warm regards,
>
> The Authors of Submission 6734.

---

> ### Author Response · Authors · 2024-11-28
> **The updated PDF has been uploaded**
>
> Dear Reviewer rah7,
>
> As the discussion deadline approaches, we have meticulously addressed your feedback and incorporated these insights into the revised version of our paper. We are confident that this updated manuscript comprehensively addresses all of your concerns.
>
> We sincerely hope that you can take a moment to check out our latest manuscript. Thank you once again for your expertise.
>
> Best regards,
>
> All authors

---

> > ### Comment · Reviewer_rah7 · 2024-12-02
> >
> > I want to thank the authors once more for their response. The principal concern from my previous reply, about your new operator being insufficiently justified, remains. It seems to me that your paper needs substantial reformulation to accommodate and motivate this new operator introduced during the rebuttal period. I therefore choose to maintain my original score, but want to encourage you to develop the motivation of your new operator further and to keep submitting this work to future conferences.

---

### Comment · Area_Chair_EKgt · 2024-11-23
**Reminder: Please Review Author Responses**

Dear Reviewers,

As the discussion period is coming to a close, please take a moment to review the authors’ responses if you haven’t done so already. Even if you decide not to update your evaluation, kindly confirm that you have reviewed the responses and that they do not change your assessment.

Thank you for your time and effort!

Best regards,
AC

---

### Author Response · Authors · 2024-11-24
**A Summary of Our Contributions and Revisions**

Dear Reviewers and ACs,

We thank the reviewers for having taken the time to read our work and for the fruitful questions and comments. We truly believe that they have helped to strengthen the paper. We are particularly happy to see that the paper’s quality has been appreciated by all reviewers.

In this global response, we aim to clarify the contributions of our work, and summarize the list of improvements we have made to the submission.

**Contributions:**

* **[High-level insight]** We identify that traditional digraph neural networks generally capture unidirectional relationships but fail to encode the asymmetric mutual path dependencies that are characteristic of digraphs. This perspective has been recognized as ```interesting ``` (*R WWVp*), ```novel ``` (*R EWXV* and *m8Kb*), ```important``` (*R EWXV*), and ```sensible ``` (*R WWVP*). This paper also provide a ```clever insight``` (*R WWVp*) and ```significant potential``` (*R m8Kb*) for advancing representation learning on digraphs.
* **[Efficient and effective model]** We introduce the novel use of commute times to quantify the strength of node-wise mutual path dependencies, an approach described as ```an intriguing idea ``` (*R WWVp*). o calculate commute times in directed graphs, we have developed a new digraph Laplacian, which is recognized for offering ```a fresh perspective with substantial potential for future research and applications``` (*R m8Kb*). Our graph rewiring method is ```simple``` and ```nice``` (*R rah7*) to compute the deterministic commute times. Building on this foundation, we propose the CGNN model, which leverages commute times to weight neighbors during message passing, a strategy deemed ```reasonable``` (*R WWVp*).
* **[Comprehensive experiments]** CGNN is supported by ```strong``` (*R rah7*) analysis of the application scope, ``solid`` and ```convincing``` (*R WWVp*) comparison with prior work, ```comprehensive``` (*R m8Kb*) component analysis, and ```interesting``` (*R rah7*) ablation study.

**The following updates have been made to the paper pdf:**

* **Refinement of the Digraph Laplacian $\texttt{DiLap}$:** We have meticulously re-derived the directed graph Laplacian matrix, $\texttt{DiLap}$, grounded in the principles of directed graph signal processing. Additionally, we have updated the experimental results to accurately reflect the impact of these methodological enhancements.
* **New theorem:** We have introduced a new theorem, complete with rigorous proof, to demonstrate that $\texttt{DiLap}$ is permutation invariant. This further substantiates the rationality of our model.
* **Polish of the writing:** Considering the valuable suggestions from all reviewers, we have thoroughly polished the writing and restructured our manuscript to enhance clarity and coherence.
* **Additional experiments:**  In response to reviewer *rah7*'s suggestions, we have included an ablation study in Appendix D.3 that examines label similarity in homophilic graphs and explores various propagation operators. Additionally, following guidance from reviewers *WWVp* and *m8Kb*, we have detailed the construction of the synthetic dataset along with associated experiments and analyses in Appendix D.4.

All modifications have been highlighted in blue in our revised manuscript.  Thanks again for your efforts in reviewing our work, and we hope our responses can address any concerns about this work.

The Authors of Submission 6734.

---

### Note · Authors · 2025-02-04

I have read and agree with the venue's withdrawal policy on behalf of myself and my co-authors.